# Principles of operation of a cerebellar learning circuit

**David J Herzfeld\*, Nathan J Hall, Marios Tringides, Stephen G Lisberger**

Department of Neurobiology, Duke University School of Medicine, Durham, United States

**Abstract** We provide behavioral evidence using monkey smooth pursuit eye movements for four principles of cerebellar learning. Using a circuit-level model of the cerebellum, we link behavioral data to learning's neural implementation. The four principles are: (1) early, fast, acquisition driven by climbing fiber inputs to the cerebellar cortex, with poor retention; (2) learned responses of Purkinje cells guide transfer of learning from the cerebellar cortex to the deep cerebellar nucleus, with excellent retention; (3) functionally different neural signals are subject to learning in the cerebellar cortex versus the deep cerebellar nuclei; and (4) negative feedback from the cerebellum to the inferior olive reduces the magnitude of the teaching signal in climbing fibers and limits learning. Our circuit-level model, based on these four principles, explains behavioral data obtained by strategically manipulating the signals responsible for acquisition and recall of direction learning in smooth pursuit eye movements across multiple timescales.

## Introduction

A hallmark of brain function is the ability to learn and remember. We can learn and successfully recall faces, events, language, concepts, places, facts, things that were frightening or rewarding, and the motor commands required to skillfully move our motor effectors. The basic currency of learning and memory is 'plasticity', changes in either the strength of synapses or the intrinsic excitability of a neuron's membrane. While decades of research have identified the basic rules that govern plasticity and, for some memory systems, have even defined the neural sites that undergo plasticity, behavioral learning is not solely a property of synapses, neurons, or individual brain sites. Rather, it is the emergent property of a complete learning neural circuit in which the sites and plasticity mechanisms of learning are embedded. Only when we incorporate our current knowledge about the specific sites of learning and the rules of plasticity with an understanding of circuit-level interactions can we truly understand learning and memory.

Here, our goal is to define the requisite set of circuit-level computational principles that operate during cerebellar-dependent motor learning. Arguably, motor learning is the domain that affords the best chance of understanding the principles of learning and memory, due to the exquisite relationship between sensory stimuli and adaptive motor behavior, combined with the tight link from known neural circuits to the output motoneurons. Across a wide range of movement modalities, motor learning depends crucially on the cerebellum (*Gilbert and Thach, 1977*; *Golla et al., 2008*; *Herzfeld et al., 2014a*; *Martin et al., 1996*; *Medina and Lisberger, 2008*; *Smith and Shadmehr, 2005*; *Robinson, 1974*), providing a well-defined neural substrate to investigate how the fundamental principles of learning are implemented in a specific neural circuit.

In the motor learning paradigm we study, direction learning in smooth pursuit eye movements, neurophysiological data has demonstrated that short-term motor learning, on the order of a single learning trial, occurs at or upstream of Purkinje cells in the floccular complex of the cerebellum (*Medina and Lisberger, 2008*; *Yang and Lisberger, 2013*; *Yang and Lisberger, 2014*). Single-trial learning is tightly linked to climbing fiber inputs, in agreement with the predictions of the classical

**\*For correspondence:**
david.herzfeld@duke.edu

**Competing interests:** The authors declare that no competing interests exist.

**eLife digest** The human brain can do many things, from reading and remembering the words written on a page to adapting and improving movements. When a movement misses its goal, the strength of the connections between cells in a part of the brain known as the cerebellum changes. The cerebellum is important for coordinating movements, including eye movements. When the connections between the cells in the cerebellum – known as neurons – strengthen or weaken, the cerebellum changes how it will respond in the future, leading to more accurate movements. However, the speed of the changes in the connections and how the connections between different neurons evolve and coordinate were unknown.

Herzfeld et al. have now combined eye-tracking studies in monkeys with computer modeling based on what is known about the neural circuits in the cerebellum to learn more about the changes in these connections. Monkeys watched a moving target that would abruptly change direction. In the next movement, the eye-tracking equipment monitored how well the monkey's eyes anticipated the unexpected change in the target's direction – a form of motor learning. Using the experimental data, Herzfeld et al. produced a model that outlines general principles of how the cerebellum might manage this process. The model suggested that neurons in one region in the cerebellum, known as Purkinje cells, learn from mistakes quickly, but have poor long-term retention. If the movement is repeated, Purkinje cells teach another area of the cerebellum, the cerebellar nucleus, which takes longer to learn but has much better retention.

Although these findings are based on a simple motor learning task, they are the first step to understanding how the brain forms memories and how we might learn more complex behaviors.

theory of cerebellar learning (*Marr, 1969*; *Albus, 1971*; *Ito, 1984*). Crucially, Purkinje cells are only two synapses away from the motoneurons that drive the adaptive motor response we measure, significantly constraining the circuit architecture that implements the transformation from sensory inputs to adapted behavior across short timescales.

However, recent behavioral results have suggested that longer term pursuit direction learning, on the order of hundreds to thousands of trials, may be mediated by multiple sites and/or learning mechanisms (*Hall et al., 2018*; *Yang and Lisberger, 2010*). The existence of multiple components in pursuit learning is consistent with behavioral and computational evidence from other cerebellar-dependent motor learning behaviors (*Boyden et al., 2004*; *Ethier et al., 2008*; *Kojima et al., 2004*; *Kording et al., 2007*; *Lee and Schweighofer, 2009*; *Medina et al., 2000*; *Smith et al., 2006*; *Kassardjian et al., 2005*; *Shutoh et al., 2006*). While plasticity is possible across many synapses in the cerebellar circuit (*Carey, 2011*; *D'Angelo and De Zeeuw, 2009*; *Hansel et al., 2001*; *McElvain et al., 2010*; *Mittmann and Häusser, 2007*), a prime candidate for long-term learning is the deep cerebellar nucleus, one synapse downstream of the Purkinje cells. Convergent neurophysiological and behavioral evidence across multiple learning paradigms lends credence to role of the deep cerebellar nucleus for long-term storage of cerebellar-dependent memories (*Raymond and Medina, 2018*; *Kassardjian et al., 2005*; *Shutoh et al., 2006*; *Lee et al., 2015*; *Raymond and Lisberger, 1998*; *Popa et al., 2016*; *Broussard and Kassardjian, 2004*; *Lisberger, 1994*). Taken together, these results highlight the idea that multiple neurophysiological mechanisms within the cerebellar circuit may underlie behavioral motor learning across longer timescales.

Our goal in the present paper is to elucidate the operation of the full and well-characterized cerebellar learning circuit. We identify the properties of the signals that guide acquisition and expression of motor learning across timescales through strategic manipulations of the experimental learning conditions coupled with quantitative measures of behavior. Our results suggest four circuit-level principles that define pursuit direction learning. First, rapid acquisition of learning occurs at the parallel fiber to Purkinje cell synapse, driven by climbing fiber responses, with limited retention. Second, learned responses in Purkinje cells guide the slow acquisition of learning in the deep cerebellar nucleus, which learns slowly but has excellent retention. Third, the functional properties of the inputs that are subject to learning are different in the cerebellar cortex and deep cerebellar nuclei. Finally, the extent of learning is limited by negative feedback from the deep nucleus to the inferior olive, reducing the teaching signal that drives learning in the cerebellar cortex.

# Results

Our goal is to understand the neural-circuit and plasticity mechanisms that mediate cerebellar motor learning across multiple timescales, ranging from a single learning trial to thousands of repetitions of the same learning stimulus. Here, we use smooth pursuit direction learning in monkeys as our representative cerebellar-dependent learning behavior. The essential cerebellar circuit for pursuit is shown in *Figure 1*. Signals related to eye kinematics and image motion are relayed to granule cells in the floccular complex of the cerebellum via mossy fibers (*Lisberger and Fuchs, 1978*; *Miles et al., 1980*; *Noda, 1986*). The information is ultimately transmitted to Purkinje cells, the sole output cells of the cerebellar cortex. Projections from Purkinje cells in the cerebellar cortex, in turn, inhibit floccular target neurons (FTNs) in the vestibular nucleus (*Lisberger et al., 1994*). Crucially, FTNs are a single synapse from the motoneurons that drive adapted behavior (*Highstein, 1973*; *Scudder and Fuchs, 1992*).

Plasticity appears to be possible at nearly every synapse in the cerebellar circuit (for a review see *Carey, 2011*), but decades of theoretical research (*Marr, 1969*; *Albus, 1971*; *Ito, 1984*) as well as neurophysiological evidence during pursuit direction learning (*Medina and Lisberger, 2008*; *Yang and Lisberger, 2014*) and other motor learning tasks (*Herzfeld et al., 2018*; *Khilkevich et al., 2016*; *Kimpo et al., 2014*) implicate plasticity at the parallel fiber to Purkinje cell synapse as critical for single-trial changes in cerebellar output. The activity of climbing fibers, originating from the inferior olive, drives unusual electrical events called 'complex spikes' in post-synaptic Purkinje cells. Complex spikes, in turn, cause long-term depression of the parallel fibers that were active in close temporal proximity to the complex spike (*Hansel et al., 2006*; *Suvrathan et al., 2016*; *Ito and Kano, 1982*).

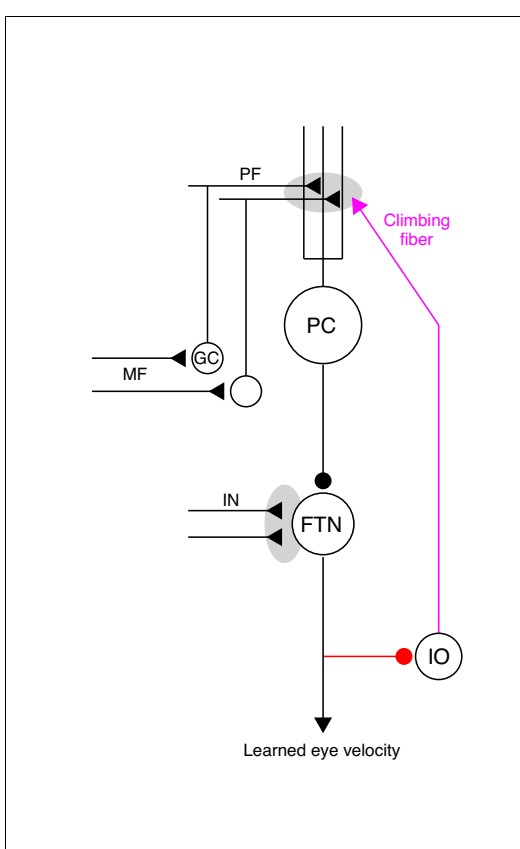

**Figure 1.** The essential cerebellar circuit responsible for the acquisition and expression of motor memories during pursuit direction learning. Mossy fibers (MFs) relay image motion and eye kinematic signals to granule cells (GCs) in the floccular complex of the cerebellum. The parallel fiber (PF) axons of cerebellar granule cells synapse on Purkinje cells (PCs). Joint activity of parallel fibers and an action potential on climbing fibers (magenta) from the inferior olive (IO) drives plasticity at the parallel fiber to Purkinje cell synapse. Purkinje cells send inhibitory projections to floccular target neurons (FTNs) in the vestibular nucleus. FTNs receive non-Purkinje cell inputs (IN). FTNs send monosynaptic projections to motoneurons, driving learned behavior. The output of the learning system sends inhibitory projections to the inferior olive (red).

The online version of this article includes the following source data for figure 1:

**Source data 1.** Figure composer source data for the cerebellar schematic.

## Characteristics of short-term pursuit learning acquisition

Given the substantial neurophysiological evidence suggesting the primary role of complex-spike-linked plasticity at the parallel fiber to Purkinje cell synapse for single-trial motor learning, our first objective was to fully characterize the properties underlying the acquisition of a motor memory following a single movement error. To isolate the properties of short-term motor learning, we developed a 'dual-trial' experimental paradigm that is a small, but important, modification of our previous methods for studying single-trial learning (see Materials and methods). In each pair of trials, the first trial is defined as the 'learning' trial. In the learning trial, the monkey begins to pursue a smoothly moving target in a randomly chosen 'pursuit direction.' After 250 ms of

target motion in the pursuit direction, the target abruptly changes direction due to the addition of a velocity component in an orthogonal 'learning direction', either +90 or −90 degrees relative to the pursuit direction (*Figure 2A*). The discrepancy between animal's smooth eye movement in the original pursuit direction and target's net direction due to the addition of orthogonal target motion creates a movement error that serves as an instruction for motor learning (*Figure 2B*). We therefore refer to the addition of the target velocity component orthogonal to the original pursuit direction as the 'instruction'.

We measure motor learning in the subsequent 'probe' trial, where the target moves in the same pursuit direction as the learning trial but without any instruction. In probe trials, the animal exhibits a 'learned response' due to the error induced by the instruction in the preceding learning trial. The learned response (*Figure 2C*, arrowhead) starts about 200 ms after the onset of target motion in the pursuit direction, anticipating the change in target direction imposed in the preceding learning trial. We measure learning in the 50 ms interval surrounding the time of the instruction, from 225 to 275 ms after the onset of pursuit target motion. The measurement interval is chosen strategically to capture the anticipatory response of the pursuit system at the time of the instruction. Later in the paper, it also allows us to measure the learned response even in learning trials, because the measurement interval precedes any visually driven eye movement resulting from retinal image motion caused by the instruction (*Hall et al., 2018*; *Yang and Lisberger, 2017*; *Medina and Lisberger, 2008*).

We first characterized the dependence of single-trial learning on the magnitude of the error imposed in the learning trial. To ensure that we were measuring only the learned response due to the occurrence of a single movement error, we prevented any long-term learning by choosing the pursuit direction for each learning-probe pair randomly from the cardinal axes (see Materials and methods). In each learning trial, we also chose the speed of the instruction randomly to be 0, 5, 10, 15, 20, 25, or 30 deg/s (*Figure 3A*), and measured the effect of varying instruction speed on the learned response in the subsequent probe trial. Single-trial learning using randomized pursuit directions creates a situation where the error magnitude experienced in the learning trial is identical to the imposed instruction magnitude, because the average eye speed in the learning direction during the learning trial is zero at the time of the instruction. This allowed us to control the size of the error that drives learning and to measure directly the shape of the relationship between learning and error.

In both monkeys, learned responses in the probe trial increased as a function of the error magnitude experienced in the learning trial

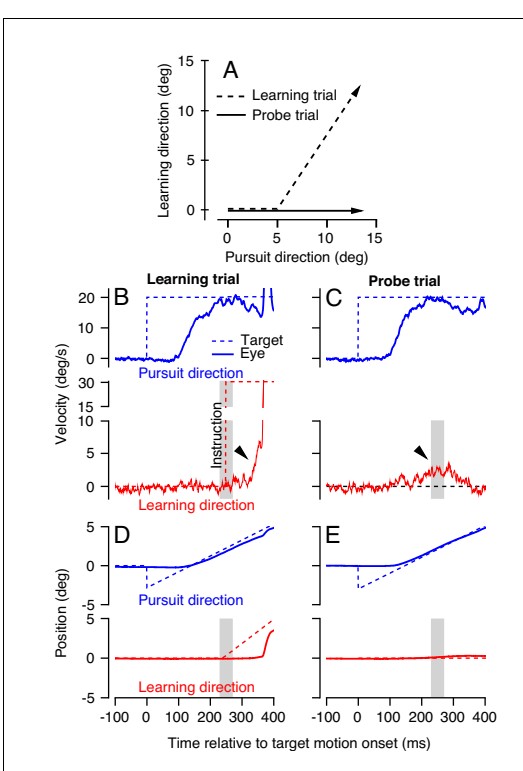

**Figure 2.** Dual-trial experimental paradigm used to selectively measure the signals associated with acquisition and generalization of a motor memory. (A) Motion of the target in two dimensions during learning and probe trials. Dashed and continuous arrows show the trajectory followed by target position during learning and subsequent probe trials, respectively. (B, C) Eye (solid) and target velocity (dashed) traces in the learning trial (B) and the probe trial (C) of each learning-probe trial pair. Blue and red traces show the pursuit direction and the orthogonal learning direction. The red dashed trace in (B) shows the onset of the instruction, which produces target motion in the learning direction at 30 deg/s. Gray shaded regions denote the time interval used to quantify behavioral learning on single trials, 225–275 ms after the onset of target motion. Arrow heads show the visually-driven response in (B) and the learned response in (C) due to presence of the instruction in the learning trial. (D, E) Eye and target position traces associated with the velocity traces in (B) and (C). The online version of this article includes the following source data for figure 2:

**Source data 1.** Figure composer source data for example learning and probe trials.

(*Figure 3B*). Quantitative analysis revealed that the learned response did not increase linearly as a function of error magnitude in the learning trial, but rather saturated as error increased (*Figure 3C*). We quantified the effect of error magnitude on single-trial learning as a sigmoid of the form:

$$Y_2(E_1) = \frac{a}{1 + e^{-\tau E_1}} - \frac{a}{2} \tag{1}$$

where $E_1$ is the magnitude of the error imposed on the learning trial and $Y_2$ is the measured learned response in the subsequent probe trial. The parameters $a$ and $\tau$ were obtained by fitting *Equation 1* to the data for each monkey in *Figure 3C* (dashed traces versus connected, filled, symbols). *Equation 1* fitted the data well for both monkeys (Monkey RE: $a$ = 1.35, $\tau$ = 0.21, $R^2$ = 0.99; Monkey YO: $a$ = 1.25, $\tau$ = 0.118, $R^2$ = 0.98). The sigmoidal model of single-trial learning was superior to an alternative linear model (Akaike's information criterion (AIC) (*Akaike, 1974*), Monkey RE: -53.3 (sigmoid) versus -14.5 (linear); Monkey YO: -29.2 (sigmoid) versus -20.5 (linear)).

Our next step was the characterize the stability of short-term pursuit learning across trials that did not include a movement error. To do so, we used a modified experimental design where the first 20 trials were standard learning trials with an instruction magnitude of 30 deg/s. The subsequent 10 trials were 'error-clamp' trials (*Figure 3D*) that we used to estimate retention in the absence of movement error (after *Scheidt et al., 2000*). Error-clamp trials used the same pursuit target motion as in the instruction trials, but the motion of the target in the learning direction was yoked to the animal's eye movement in that direction, to ensure that the animal experienced no visual error signals in the learning direction. Thus, any trial-to-trial change in the animal's learned response over the course of 10 error-clamp trials was due solely to forgetting of the previously acquired short-term motor memory. We normalized the learned response in the 10 error-clamp trials by the magnitude of the animal's learned response in the final learning trial of the 20-trial learning block.

Both monkeys demonstrated an exponential decay in the magnitude of the learned response across error-clamp trials (*Figure 3E*). Fitting an exponential model to each monkey's retention data (*Figure 3E*, black lines) suggested a retention constant of approximately 0.85 (95% confidence intervals based on four experiments per monkey, Monkey RE: 0.80–0.86, Monkey YO: 0.85–0.91). Therefore, in the absence of any error to drive learning, approximately 15% of the short-term learned response is forgotten on each trial. We performed this experiment only for instruction speeds of 30 deg/s because the larger amplitude of learning provided more reliable estimates of retention.

## Generalization of the expression of single-trial pursuit learning

We next sought to use generalization of learning to constrain the functional properties of the site of plasticity that causes single-trial motor learning. Our strategy is based on the following line of reasoning. Imagine, for example, that the input signals that are subject to single-trial learning are linearly related to eye speed in the pursuit direction ($v_e$) and that plasticity operates as a scalar gain ($g$). Then, the learned change in firing of the relevant post-synaptic cells should be proportional to $gv_e$. As a result, the learned eye speed should generalize linearly as a function of pursuit speed in the probe trial. We can invert this logic to use the measured characteristics of generalization of learning to reveal the functional properties of signals that undergo plasticity. Under the assumption that the site of plasticity for single-trial learning acts as a scalar gain, if learned eye speed generalizes linearly with pursuit eye speed in the probe trial then the population response of neurons upstream from the site of plasticity should be linearly related to eye speed in the pursuit direction.

We again employed our dual-trial paradigm with a randomized pursuit direction for each pair of learning and probe trials. Here, the learning trials always had a pursuit speed of 20 deg/s and an orthogonal instruction speed of 30 deg/s. The pursuit speed of the target in the probe trial was selected randomly from 5, 10, 15, or 20 deg/s (*Figure 4A*).

We found that average eye speed in the learning direction scaled with the speed of the probe target for both monkeys RE and YO, especially in the interval from 225 to 275 ms after the onset of target motion (*Figure 4B*, shaded area). We note that scaling is less clear earlier in the probe trial for Monkey YO, partly because the animal's eye speed in the pursuit direction scales poorly with target speed during this earlier interval. A RM-ANOVA showed a main effect of eye speed in the pursuit direction in the probe trial on the amount of measured behavioral learning for both monkeys (Monkey RE: $F_{(3, 6)} = 24.3$, $p < 10^{-3}$, Monkey YO: $F_{(3, 15)} = 13.7$, $p < 0.001$). For both monkeys, the effect of eye speed in the pursuit direction on the learned response from 225 to 275 ms was highly

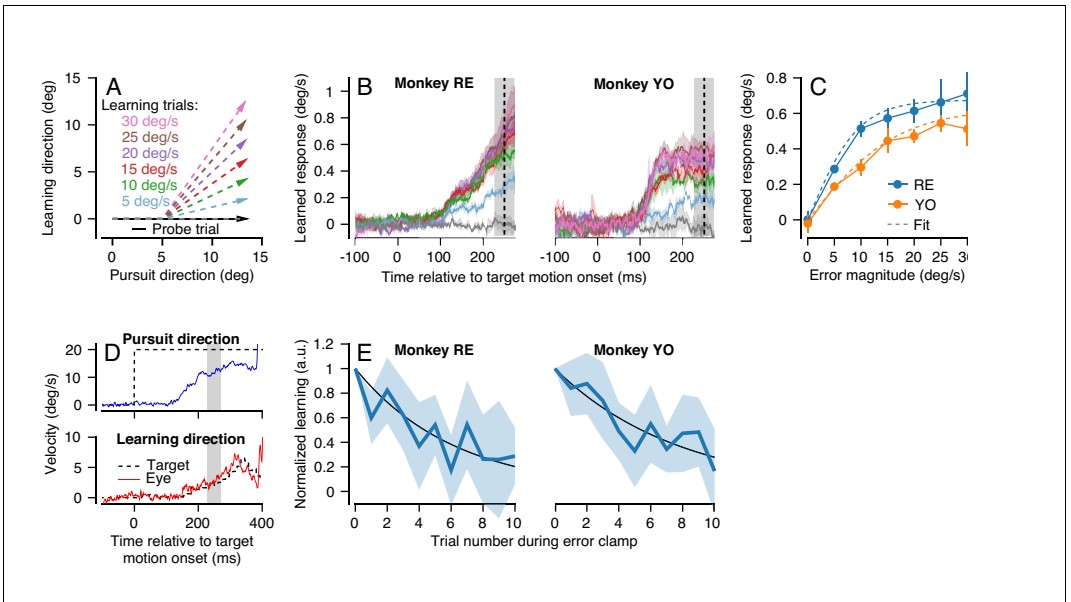

**Figure 3.** Characteristics of short-term pursuit learning acquisition and retention. (**A**) Colored arrows show the target position trajectory in two dimensions during learning trials. Different colors show learning trials with instruction speeds of 5, 10, 15, 20, 25, or 30 deg/s. Black arrow shows the subsequent probe trial (learning speed 0 deg/s). Pursuit and instruction directions were randomized from trial-to-trial (see Materials and methods). (**B**) Average eye velocity in the direction of the previous instruction (learning direction) as a function of time during probe trials averaged across experiments from Monkeys RE (left) and YO (right). Different colors indicate average responses for the different error magnitudes using the same color scheme as in (**A**). The vertical dashed line shows the time of the instruction in the preceding learning trial and the gray shaded region shows the interval used for quantitative analysis, 225 to 275 ms after the onset of pursuit target motion. (**C**) The amplitude of single-trial learning as a function of the error magnitude created by the previous instruction. Dashed lines show best-fit sigmoid for each monkey. (**D**) Example 'error-clamp' trial used to minimize the retinal image motion in the learning direction, allowing characterization of motor memory retention in the absence of error. Red and blue traces show eye velocity and the black dashed traces show target velocity. (**E**) Trial-course of forgetting after 20 learning trials, probed under error-clamp conditions for Monkeys RE (left) and YO (right). Data are normalized to the magnitude of behavioral learning in the last learning trial (trial 0). Black curves show best fitting exponentials. Error bands and error bars show ± SEM across experiments for each monkey.

The online version of this article includes the following source data for figure 3:

**Source data 1.** Figure composer source data characterizing single-trial pursuit learning.

---

linear (*Figure 4C*, Monkey RE: $R^2 = 0.99$, Monkey YO: $R^2 = 0.97$). These results suggest that signals related to the speed of eye motion in the pursuit direction may be part of the substrate that is subject to plasticity. The linear generalization of single-trial learning with pursuit direction eye speed is consistent with prior observations that the eye speed at the time of the instruction linearly modulates the expression of behavioral learning (*Hall et al., 2018*).

In contrast, target/eye speed in the pursuit direction during the *learning* trial did not affect the expression of single-trial learning on the subsequent probe trial. Here, we again used the dual-trial paradigm with pursuit speed randomly chosen to be 5, 10, 15, or 20 deg/s in the learning trials but always 20 deg/s in the probe trials. The instruction occurred 250 ms after the onset of target motion and had a magnitude of 30 deg/s. The direction of the instruction was chosen randomly to be either +90˚ or −90˚ relative to the pursuit direction. Both Monkeys RE and YO (*Figure 4D*) showed no effect of pursuit speed in the learning trial on the magnitude of the learned response measured in the probe trials. A RM-ANOVA failed to show a significant effect at 250 ms in the probe trial (*Figure 4E*, Monkey RE: $F_{(3, 15)}=0.56$, $p=0.65$, Monkey YO: $F_{(3, 15)}=0.39$, $p=0.76$). Thus, we can conclude that target/eye speed in the pursuit direction has distinct and very different effects on the acquisition of a motor memory during the learning trial versus on the expression of learning in the probe trial.

## Cerebellar model of short-term motor learning

Previous neurophysiological results suggest that complex-spike-linked plasticity at the parallel fiber to Purkinje cell synapse in the cerebellar cortex is likely the primary driver of single-trial pursuit learning (*Medina and Lisberger, 2008*; *Yang and Lisberger, 2014*). Combining these neurophysiological observations with our behavioral results thus far allows us to begin to constrain the parameters of a cerebellar model of motor learning.

A model of short-term pursuit learning appears in *Figure 5A* (*Source code 1*). Here, we chose to model the activity of parallel fibers as linearly related to eye speed in the pursuit direction, to account for the generalization of single-trial learning (*Figure 4B–D*). We use only two parallel fibers with opposite preferred directions (more parallel fibers are trivially possible but unnecessary) with an arbitrary scaling parameter, $r$, that converts eye velocity to spikes/s:

$$PF^1 \propto r\dot{E}$$
$$PF^2 \propto -r\dot{E}$$

(2)

In the absence of learning, the parallel fiber inputs cancel perfectly, ensuring that the firing of the post-synaptic Purkinje cell does not modulate in relation to eye movement in the pursuit direction, consistent with experimental data (*Medina and Lisberger, 2008*). We can model Purkinje cell firing on the $n^{th}$ trial, $PC_n$, as the weighted contributions of the pre-synaptic parallel fibers and a background response, $PC_0$:

$$PC_n = w_n^1 PF_n^1 + w_n^2 PF_n^2 + PC_0$$

(3)

The synaptic weight for each parallel fiber, $w^j$, is adaptable via the complex spikes caused by climbing fiber inputs. The plasticity rule includes two terms: (1) a decay term, $\alpha_{PF}$, that allows the relaxation of any learned weight changes back to their baseline value in the absence of error and (2) an update term that depresses weights by a fractional amount (β) depending on the positive firing on the parallel fiber in question and the probability of a complex spike given the neural representation of the image motion from the instruction, $P(CS|E_n)$:

$$w_{n+1}^i = \begin{cases} w_n^i - (1-\alpha_{PF})(w_n^i - w_0^i) - \beta P(CS|E_n) & \text{if } PF^i > 0 \\ w_n^i - (1-\alpha_{PF})(w_n^i - w_0^i) & \text{if } PF^i \leq 0 \end{cases}$$

(4)

When $\alpha_{PF}=1$, the weights are maintained completely from one trial to the next and learning is allowed to perfectly accumulate; when $\alpha_{PF}<1$, the weights decay back to their baseline levels, $w_0^i$, in the absence of any climbing fiber input. We model only learning for instructions in the direction that causes complex spikes so that we do not need to explicitly simulate synaptic potentiation beyond the relaxation back toward the baseline weights provided by $\alpha_{PF}$.

Our results in *Figures 3* and *4* suggest appropriate values for the unknown parameters of this simplified cerebellar model of learning. First, we know that, in the absence of error, short-term motor learning decays across trials. The time constant analysis in *Figure 3E* suggests that $\alpha_{PF} \approx 0.85$ for both monkeys. In addition, *Equation 1* describes how the parallel fiber to Purkinje cell weights are modified as a function of error during single-trial learning: $Y_2(E_1) \propto \beta P(CS|E_1)$. Previous neurophysiological data has demonstrated that the probability of observing a complex spike following a 30 deg/s instruction is approximately 30% (*Yang and Lisberger, 2014*; *Yang and Lisberger, 2017*). Given these observations, we can derive a rule linking the probability of observing a complex spike to a given error magnitude using a sigmoid function with the same form as *Equation 1*:

$$P(CS|E_n) = \frac{0.6}{1 + e^{-\tau E_n}} - 0.3$$

(5)

Here, the probability of observing a complex spike saturates at 30% for large error magnitudes. The value of τ matches the values we obtained for each monkey in *Equation 1*.

Purkinje cells inhibit the floccular target neurons (FTNs) in the vestibular nucleus which, in turn, modulate eye movements via their projections to motoneurons. In the absence of other inputs to FTNs that are modulated by learning, the change in the firing of FTNs from their baseline activity is $FTN_n = PC_0 - PC_n$. The resulting learned response, $Y_n$, measured via generated eye movements is then $Y_n = cFTN_n$, where $c$ is an arbitrary constant that scales FTN activity to deg/s.

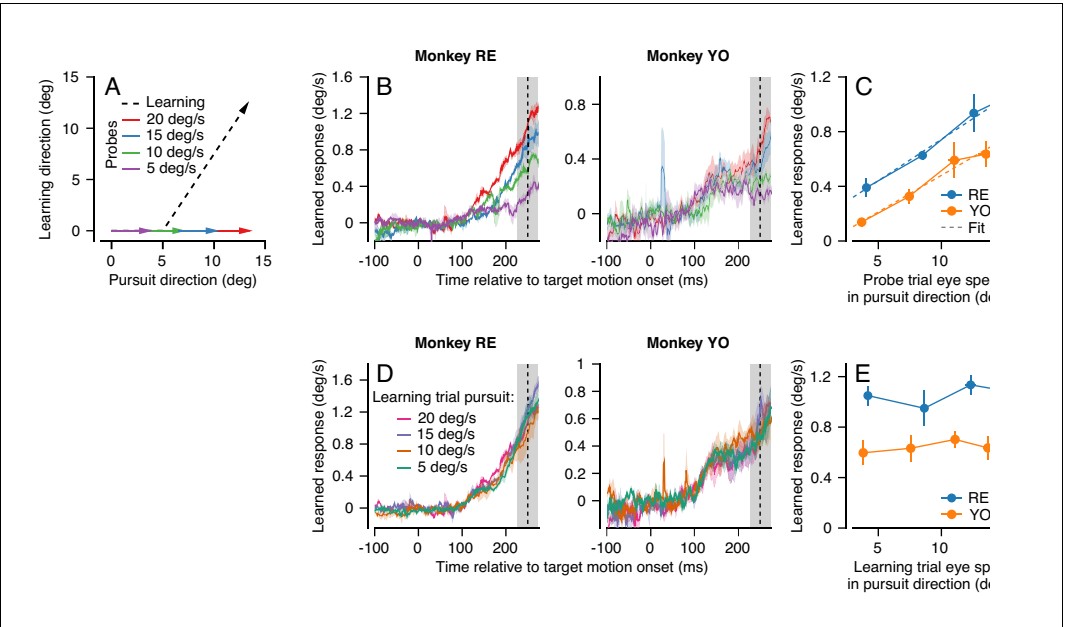

**Figure 4.** Effect (and non-effect) of pursuit eye speed on expression (and acquisition) of single-trial learning. (A) Experimental paradigm for evaluating the effect of pursuit speed on expression of learning, showing target motion in two dimensions. Black dashed trace shows the learning trial and colored arrows show probe trials with different pursuit speeds. (B) Learned eye velocity in the direction of the previous instruction (learning direction), recorded in the probe trials and averaged across experiments for each monkey. Different colors show responses for different pursuit speeds in the probe trial, as in (A). Vertical dashed line shows the time of the instruction from the preceding learning trial. (C) Symbols connected by lines show the learned response measured in the gray shaded region in (B) as a function of the measured eye velocity in the pursuit direction during the same interval. Dashed lines show the best fit linear model. (D) Learned eye velocity in the direction of the previous instruction, recorded in the probe trial (20 deg/s pursuit speed) and averaged across experiments for each monkey. Different colors show responses for different pursuit speeds in the preceding learning trial. Vertical dashed line shows the time of the instruction in the preceding learning trial. (E) Summary of the learned response in the probe trial during the gray-shaded region in (D) as a function of the eye speed in the pursuit direction in the preceding learning trial. Error bands and error bars show ± SEM across experiments for each monkey.
The online version of this article includes the following source data for figure 4:

**Source data 1.** Figure composer source data showing the effects of pursuit speed on expression of single-trial learning.

*Equations 2-5* describe a 'single-learning-process' model. The constants, $r$, $\beta$, $w_0^i$ and $c$ are arbitrary and affect only the scaling of firing rates to eye movements. The values we chose for these parameters are shown in *Table 1*, but any number of parameter combinations would give qualitatively similar behavioral results.

The single-learning-process model appropriately estimated our behavioral observations for two different stimulus conditions, one that was more-or-less built into the model and one that matches our data as an emergent property of the model. First, the trial-to-trial change in output following a range of imposed error magnitudes closely matched the learned behavior of both animals (*Figure 5B*), because this relationship was built into the model based on the data in *Figure 3C*. Second, the model reproduced the results of an experiment that was a superset of the conditions used in *Figure 4A–C*. Here, we presented a series of dual-trial stimuli where the instruction trial used a randomly chosen pursuit speed of 0, 10, 15, or 20 deg/s and delivered an instruction at 30 deg/s. The probe trial also delivered a pursuit speed that was randomly chosen from 0, 10, 15 or 20 deg/s. For each probe trial, we computed the 'learning expression ratio' as the magnitude of the learned response expressed in each probe trial normalized by the average learning expressed in probe trials with the same pursuit speed as the preceding learning trial. When the pursuit speed in probe trial matched the pursuit speed in the learning trial, the learning expression ratio is ~1.0. We then plotted

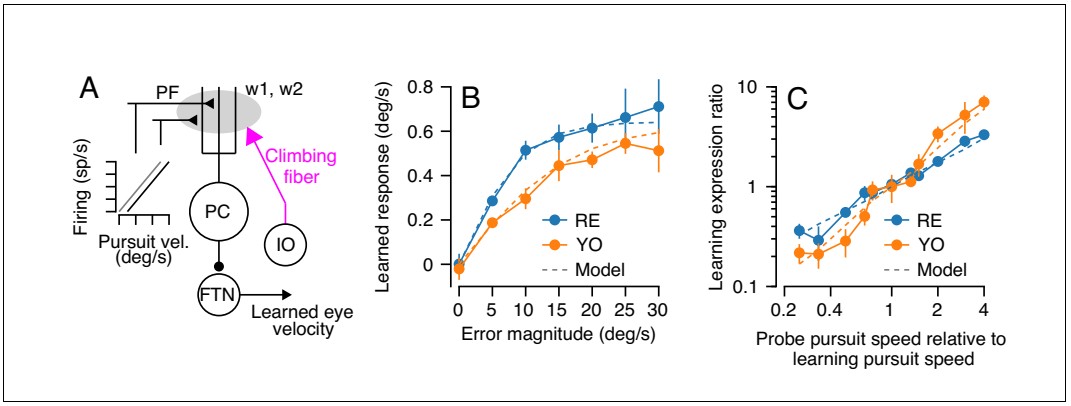

**Figure 5.** Properties of a cerebellar model of single-trial learning with a single site of plasticity. (A) Simplified model of cerebellar learning with a single-site of plasticity at the parallel fiber to Purkinje cell synapse (gray shaded oval). Abbreviations are the same as in *Figure 1*. In this single-learning-process model, the firing of parallel fibers is linearly related to the speed of the eye in the pursuit direction. (B) Dashed lines show the model's prediction of learned responses as a function of error magnitude. Filled symbols connected by lines are reproduced from *Figure 3C*. (C) Learning expression ratio as a function of the ratio of pursuit speed in the probe trial to the pursuit speed in the learning trial (note that both axes are on a log scale). By definition, the learning expression ratio is ~1.0 when the pursuit speed in the probe trial and the preceding learning trial are identical. As in (B), dashed lines show the prediction of the single-learning-process model and symbols connected by lines summarize data from the two monkeys. Error bars show ± SEM across experiments for each monkey.

The online version of this article includes the following source data for figure 5:

**Source data 1.** Figure composer source data comparing the single-site model of cerebellar learning with behavioral results.

the learning expression ratio as a function of the ratio of the pursuit speeds of the probe trial and the preceding learning trial to assess the generalization function of learning across any combination of pursuit speeds in the learning and probe trials. The model reproduces the results of this experiment on both monkeys (*Figure 5C*), demonstrating strong linear generalization across probe pursuit speeds.

In summary, a cerebellar model with a single learning site where plasticity operates on signals linearly related to pursuit eye speed predicts all of our behavioral data for single-trial learning.

## Long-term motor learning is not the accumulation of single-trial learning

In the more natural situation where the same instruction for learning occurs on movement after movement and is corrected gradually, learning could be simply due to an accumulation of single-trial

**Table 1.** Pursuit learning model parameters.

| Model parameter | Value | Description |
| --- | --- | --- |
| $r$ | 1 | Parallel fiber eye velocity (deg/s) to firing rate (Hz) |
| $w_0$ | 1 | Parallel fiber to Purkinje cell baseline weights |
| $\alpha_{PF}$ | 0.85 | Parallel fiber to Purkinje cell weight retention |
| $\beta$ | 1.5 | Change in parallel fiber to PC weights due to a complex spike |
| $c$ | 0.0625 | FTN firing rate (Hz) to eye movement (deg/s) |
| $\sigma$ | 7.5 | Standard deviation of Gaussian FTN input responses |

learning plus forgetting at the rate of 15% per trial. At some point, the forgetting and learning on each trial would balance and the learning curve would reach an asymptote. To provide a dataset to ask whether the 'single-learning-process' model of single-trial cerebellar learning with forgetting (*Equations 2-5)* can explain the accumulation of behavioral learning across multiple learning trials, monkeys completed 100 consecutive learning trials. Each learning trial used identical pursuit and learning directions and speeds. Within each learning block of 100 trials, the instruction magnitude was either 0, 5, 10, 15, 20, 25, or 30 deg/s. Note that during 100-trial learning blocks, we control the magnitude of the instruction, but the magnitude of the error on any given trial is the difference between the instruction and the monkey's expression of behavioral learning on that trial. When the same learning trial is repeated, acquisition of behavioral learning leads to a gradual reduction in the magnitude of the error for a given magnitude of instruction. Therefore, we use the terms 'instruction magnitude' and 'error magnitude' carefully to convey this distinction.

For each instruction magnitude, the trial-course of learning revealed dual-exponential learning curves that largely saturated by the end of 100 learning trials (*Figure 6A*). Yet, even after 100 consecutive learning trials, the magnitude of the animal's learned response was always much smaller than the speed of the imposed instruction: neither animal ever exhibited full compensation for the imposed instruction (*Figure 6B*). It is especially noteworthy that the system is able to achieve a learned response of 5 deg/s after 100 learning trials for an instruction at 30 deg/s, but not for an instruction at 5 deg/s.

The effect of instruction magnitude on the learned response was much more linear after 100 consecutive learning trials than after a single learning trial. The learned eye speed in the last 25 trials of the learning epoch was linearly related to instruction magnitude for both monkeys (Monkey RE: $R^2 = 0.99$, t-test on the slope of the regression fit: t(5)=21.9, $p<10^{-4}$; Monkey YO: $R^2 = 0.99$, t(5) =18.9, $p<10^{-4}$). After 100 learning trials, a linear model was preferred when compared against the best-fit saturating model from *Equation 1* (AIC, Monkey RE: 7.7 (sigmoid) versus 1.5 (linear); Monkey YO: 0.6 (sigmoid) versus 0.02 (linear), *Figure 6C*). The difference in the effect of instruction magnitude on learning after 1 versus 100 learning trials hinted that multiple plasticity mechanisms and/or sites within the cerebellar circuit could be driving learning across the different trial-courses.

Using the single-learning-process model, we simulated blocks of 100 consecutive learning trials. In each trial, we drove learning with an error that was the difference between model output and $I_n$, the imposed instruction: $E_n \equiv I_n - cFTN$, simulating the gradual reduction in the error signal for a fixed instruction magnitude due to the acquisition of learning across trials.

The single-learning-process model failed to predict two crucial features of the true trial-course of learning. First, it produced single exponential trial-courses that learned too quickly and reached asymptote within the first 20 learning trials (*Figure 7A*). Second, it systematically over-estimated the extent of asymptotic learning for instruction magnitudes between 5 and 20 deg/s (*Figure 7B*, arrows) and underestimated learning for instruction magnitudes of 30 deg/s. Indeed, the model predicted a sigmoidal relationship between asymptotic learning and instruction magnitude (*Figure 7B*), inherited from the relationship between single-trial learning and instruction magnitude, in contrast to the highly linear relationship in the data (*Figures 6C* and *7B*). We note that we did not fit the model to the data. Instead, we used the parameters derived from our behavioral experiments that allowed *Equations 2-5* to predict the details of single-trial learning. We therefore conclude that pursuit learning cannot be mediated solely by the accumulation of single-trial learning with forgetting (*Yang and Lisberger, 2010*; *Hall et al., 2018*).

## A cerebellar circuit model that predicts behavioral learning across timescales

Our next step was to identify the minimal principles of a cerebellar circuit model that can explain behavioral learning across multiple timescales. The features of behavioral learning in *Figures 3* and *4* mandate several properties of a more complete cerebellar model: (1) acquisition of single-trial learning saturates as a function of the magnitude of the error according to a sigmoidal relationship dictated by the data in *Figure 3C*; (2) at least across the 100 learning trials we tested in *Figure 6* but also after 2000 trials (*Hall et al., 2018*), the asymptotic response for a consistent instruction magnitude falls short of completely correcting the movement error created by the imposed instruction; (3) the trial-course of learning across long bouts of consecutive learning trials (*Hall et al., 2018*) and the different relationship between learned response and error magnitude after 1 versus 100

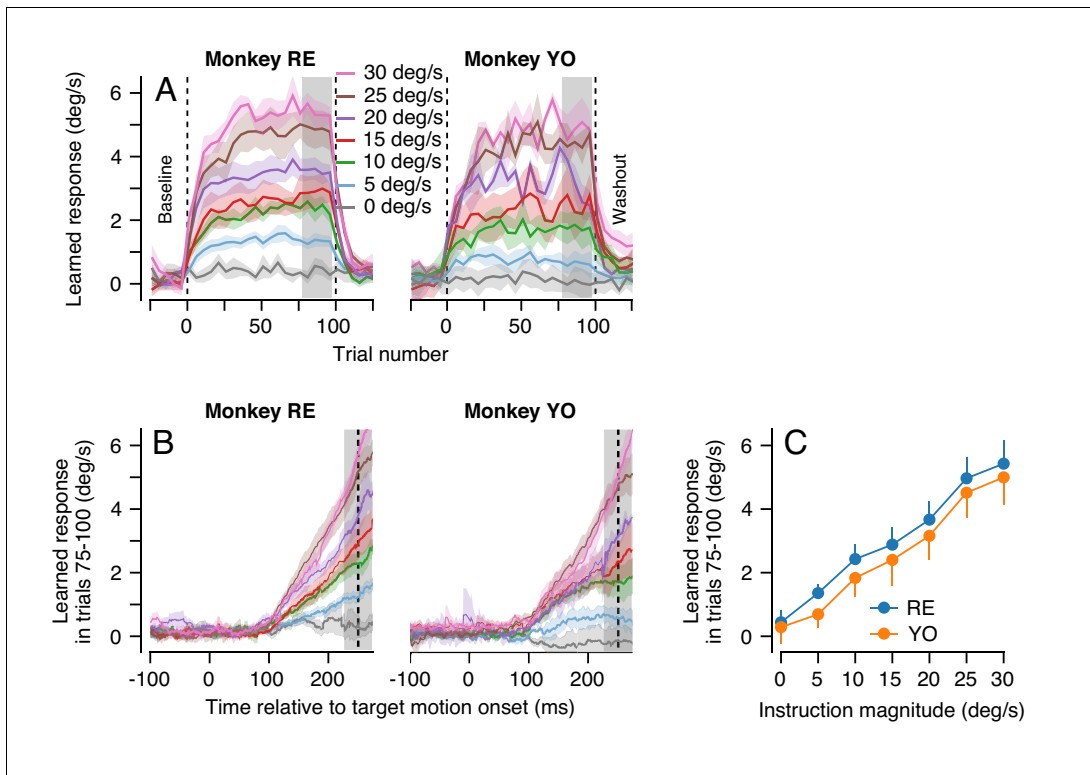

**Figure 6.** Effect of instruction magnitude on behavior learning after 100 trials of repeated instructions. (**A**) Trial-courses of learning for blocks of 100 learning trials with a consistent instruction magnitude. Different colors show trial-courses for different instruction magnitudes. Dashed vertical lines show the first and last learning trials. Data are binned in discrete five-trial bins and averaged across experiments in each monkey. Gray shading shows the range of trials used to assess the magnitude of asymptotic learning. (**B**) Average learned eye velocity as a function of time in the final 25 learning trials for blocks of 100 learning trials, averaged across experiments for each monkey. Gray shading shows the time interval used for quantitative analysis of the learned eye velocity. (**C**) Summary of the effect of instruction magnitude on the asymptotic learned eye velocity in the final 25 trials of a learning block. Error bands and error bars show ± SEM for each monkey across repetitions of 100-trial learning blocks.

The online version of this article includes the following source data for figure 6:

**Source data 1.** Figure composer source data showing the cumulative effects of multiple learning trials.

trials suggest the presence of at least two learning processes (*Hall et al., 2018*; *Kojima et al., 2004*; *Lee and Schweighofer, 2009*; *Smith et al., 2006*).

To test our conclusion about multiple learning processes contributing to cerebellar learning across longer learning sessions, we next asked whether the retention of the motor memory changed across long bouts of learning trials. We used the same error-clamp strategy as before, but now to measure the retention constant after blocks of >1000 identical learning trials. At the end of long blocks of learning trials, we alternated short blocks of 10 error-clamp trials with runs of 20 learning trials to 'top up' the asymptotic learning. We found significantly better retention after 1000 trials than after 20 trials (*Figure 8A*): the measured retention constant after 1000 trials was approximately 0.95 (95% confidence intervals, Monkey RE: 0.95–0.97, Monkey YO: 0.94–0.96)). These data suggested that the motor memory gradually transitions, at least conceptually, from a relatively labile, low-retention learning process early in learning to a second learning process with high retention (*Smith et al., 2006*; *Herzfeld et al., 2014a*).

Armed with knowledge that learning appears to transition from a low-retention learning process to a higher retention process after extended learning, we expanded our circuit model of cerebellar learning to include a secondary site of plasticity (*Figure 8B*, *Source code 2*). Given neurophysiological results from other cerebellar-dependent learning tasks (*Kassardjian et al., 2005*; *Shutoh et al., 2006*; *Lee et al., 2015*; *Raymond and Lisberger, 1998*; *Lisberger, 1994*) as well as the proximity of

the cerebellar circuit to motoneurons, we view the floccular target neurons, or FTNs, in the vestibular nucleus as the most likely candidate for long-term storage of learning. FTNs are immediately downstream of the Purkinje cells, the presumed site of rapid learning, and synapse onto output motoneurons.

We modeled the learned responses of FTNs on trial $n$ as the sum of the firing of input axons, $IN_n^i$, weighted by the scalar synaptic weight, $v_n^i$, minus the baseline-subtracted firing of the Purkinje cell:

$$FTN_n = \sum_i v_n^i IN_n^i - (PC_n - PC_0) \tag{6}$$

To account for a secondary site of plasticity within the learning circuit, we implemented Hebbian-style plasticity at the synapse from the eye movement inputs to FTNs. Plasticity at the inputs to FTNs was driven by the learned responses of presynaptic Purkinje cells:

$$v_{n+1}^i = v_n^i + \eta(PC_0 - PC_n)IN_n^i \tag{7}$$

Here, $\eta$ determines the strength of the Purkinje cell's teaching signal and the rate of plasticity in the inputs to FTNs. The value of $\eta$ was set to be small so that the FTNs represented the site of a slow-learning process, in contrast to the rapid learning at the parallel fiber to Purkinje cell synapse. Based on our behavioral results in *Figure 8A*, plasticity in the slow-learning-process is retained almost perfectly, so that *Equation 7* lacks the forgetting factor provided by $\alpha_{PF}$ in *Equation 4*. Together, *Equations 6 and 7* assume that the learning in the Purkinje cell responses acts as a teacher for long-term plasticity at the inputs to the FTNs.

Simulation of *Equations 2-7* for a variety of instruction magnitudes (*Figure 8C*) correctly demonstrates one crucial feature of our long-term learning data. The simulated learning is dual-exponential, with initial fast acquisition and a second, slower, timescale of acquisition. However, the simulations do not predict a crucial aspect of our behavioral data. Due to the stability of the long-term memory at the FTN inputs, the asymptotic response of the model is much larger than is found in our data. Also, *Figure 8C* shows that the slow-learning process produces a linear relationship between asymptotic learning and instruction magnitude, but only after 2000 trials and not after 100 trials (vertical dashed line) as seen in the data. The excessive learning in the model of *Figure 8B* suggests the existence of inhibition within the learning circuit.

Multiple deficiencies of the model in *Equations 2-7* are corrected when recurrent inhibition from the cerebellum to the inferior olive modulates the error signals used to drive learning at the parallel fiber to Purkinje cell synapse. We modeled inhibition by reducing the magnitude of the error signal that drives climbing fiber responses in proportion to the size of the learned response:

$$E_n = I_n - cFTN_n - \frac{\gamma cFTN_n}{\dot{E}} \tag{8}$$

Here, $I_n$ is the speed of the instruction on the $n^{th}$ trial and $cFTN_n$ is the FTN firing converted to units of learned eye velocity. The term $(I_n - cFTN_n)$ in *Equation 8* represents the physical error that results from the difference between the instruction magnitude and the response of FTNs. This formula makes the distinction between the physical error, $(I_n - cFTN_n)$, and the learning system's internal representation of this error used to drive learning in the cerebellar cortex ($E_n$). The last term in *Equation 8* implements modulation of the internal representation of the physical image motion in proportion to $\gamma$ and the learned response, $cFTN_n$. There is no recurrent control when the value of $\gamma$ is zero. The modulation is normalized by eye speed in the pursuit direction ($\dot{E}$) so that inhibition of the climbing fiber input occurs in proportion to the change in weight of the parallel fiber to Purkinje cell synapse. The combination of the Purkinje cell modulation as a teacher (*Equation 7*) as well as the presence of recurrent inhibition (*Equation 8*) transfers the motor memory from the cerebellar cortex to the vestibular nucleus across long blocks of learning trials. We note that the probability of complex spikes decreases considerably across 100 repetitions of the same learning trial (*Yang and Lisberger, 2017*), compatible with the mechanism implemented by *Equation 8*.

We adjusted the strength of the teaching signal from the Purkinje cells to the FTN inputs ($\eta$) and the strength of recurrent inhibition of the instructive signal ($\gamma$) to allow the model in *Figure 8B* and *Equations 2-8* to reproduce both the two-component trial-course of learning (*Figure 8D*) and the

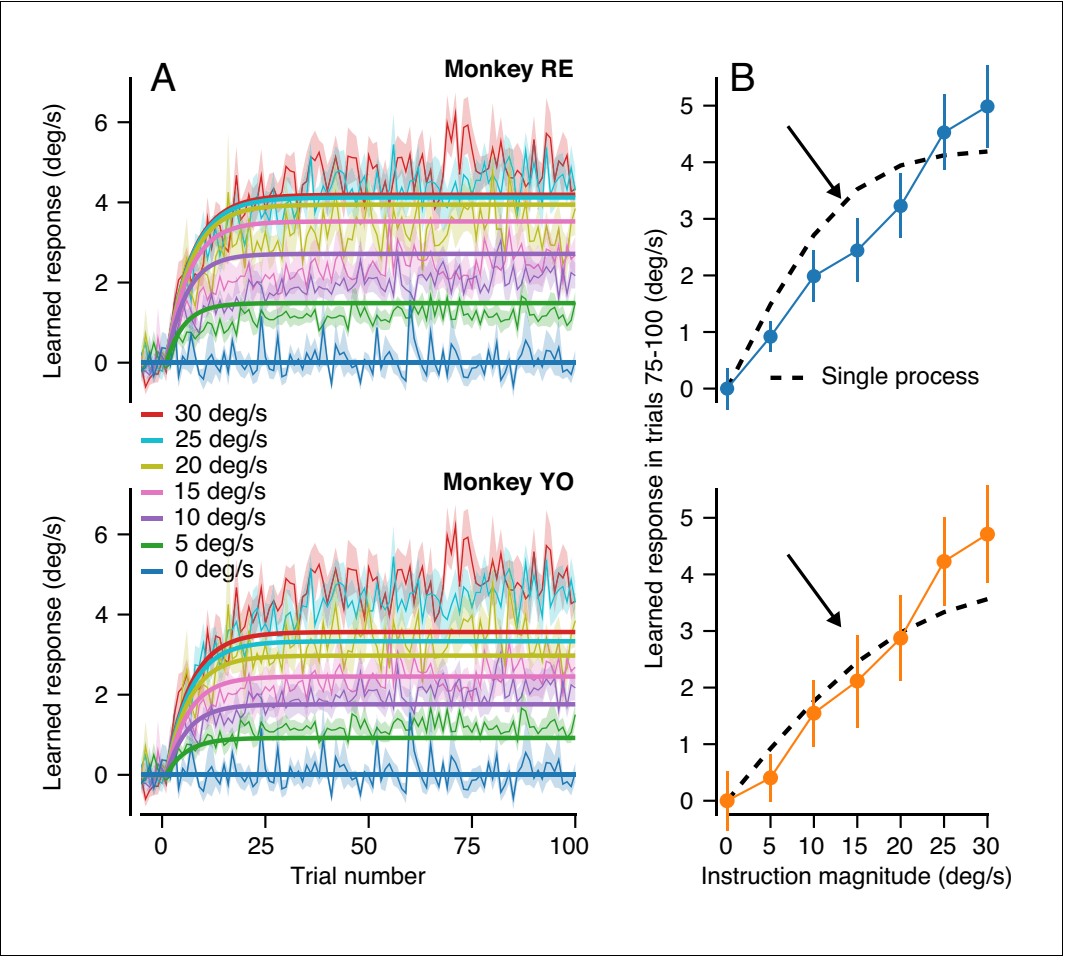

**Figure 7.** Long-term behavioral learning is not merely accumulation of single-trial learning. (A) Trial-courses of learning for different instruction magnitudes, averaged across experiments for both monkeys. Thin traces with error bands show learning as a function of trial number as mean ± SEM (data reproduced from *Figure 6A* without binning). Bold traces show the predictions of the single-learning-process model described in the text. (B) Asymptotic learning measured in the last 25 trials of a 100-trial learning block, plotted as a function of the speed of the instruction in that block. Connected symbols and error bars show the data for each monkey (reproduced from *Figure 6C*). Dashed traces show the predictions of the single-learning-process model. Error bands and error bars show ± SEM for each monkey across repetitions of 100-trial learning blocks.

The online version of this article includes the following source data for figure 7:

**Source data 1.** Figure composer source data showing the progression of learning across 100 trial blocks of learning trials compared to the results predicted by a single-site model of cerebellar learning.

---

correct linear relationship between the asymptotic learning and instruction magnitude after 100 learning trials (*Figure 8E*). The best-fit model (two unknowns, η and γ) agrees well with the data for both monkeys (*Figure 8E*, black curves, Monkey RE: $R^2$ = 0.99; Monkey YO: $R^2$ = 0.97). In addition, both monkeys had similar values for both estimated parameters (95% confidence intervals, Monkey RE: η = $5.1 \times 10^{-4}$–$9.2 \times 10^{-4}$, γ = 55.9–68.5, Monkey YO: η = $4.5 \times 10^{-4}$–$9.4 \times 10^{-4}$, γ = 38.8–65.0). Linearity emerges after 100 trials because recurrent feedback switches the mechanism driving asymptotic behavior from a tradeoff between learning and forgetting to an emergent property of the operation of a dynamic feedback loop.

The successful model in *Figure 8B* has three crucial features: (1) a site of rapid learning with limited retention at the parallel fiber to Purkinje cell synapse, (2) gradual transfer of learning from the cerebellar cortex to a slower-learning process with excellent retention at the inputs to the FTNs, and (3) recurrent inhibition from FTNs to the inferior olive that limits learning and linearizes the stimulus-

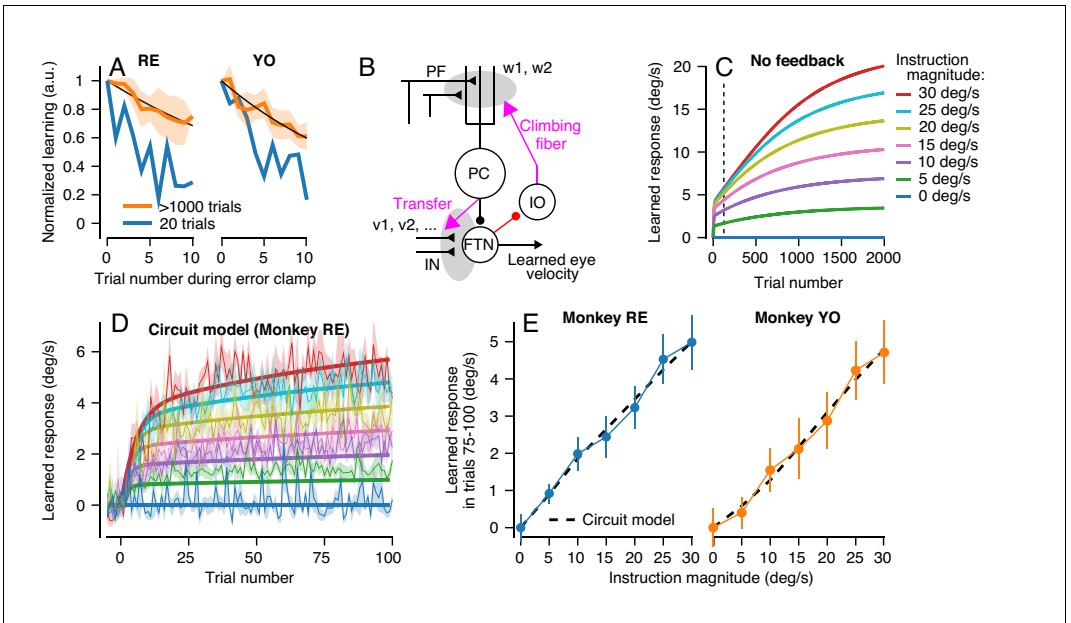

**Figure 8.** A cerebellar model that captures the amplitude and trial-course of learning across multiple timescales. (**A**) Trial course of forgetting under error-clamp conditions. Orange and blue traces show forgetting after >1000 or 20 trials (the latter reproduced from *Figure 3E*). Black curves show best fitting exponentials. Error bands show SEM across n = 4 experiments. (**B**) Schematic diagram of a cerebellar model with two sites of plasticity, capable of recapitulating the properties of pursuit learning across multiple timescales. Magenta arrows denote teaching/ transfer signals. The red pathway denotes recurrent inhibition of the internal representation of the error that drives learning. (**C**) Trial-course predictions of the cerebellar model in (**B**) without recurrent feedback to the inferior olive for a range of instruction magnitudes. (**D**) Exemplar fits of the cerebellar learning model in (**B**) to trial-courses of learning for repeated presentations of instructions at different speeds. Thin traces with error bands show learning averaged (± SEM) across multiple experiments as a function of trial number for Monkey RE (reproduced from *Figure 6A*, unbinned). Bold traces show the predictions of the best-fitting cerebellar circuit model for each instruction magnitude. (**E**) Asymptotic learning measured in the last 25 trials of a 100-trial learning block, plotted as a function of the speed of the instruction in that block. Connected symbols and error bars show the data for each monkey (reproduced from *Figure 6C*). Dashed traces show the predictions of the asymptotic response of the cerebellar model from (**B**).

The online version of this article includes the following source data for figure 8:

**Source data 1.** Figure composer source data showing the predictions of a model of cerebellar motor learning which includes multiple sites of plasticity operating at different timescales.

response function of learning by controlling the error signals available to drive learning at the parallel fiber to Purkinje cell synapse.

## Changes in the pattern of generalization over multiple timescales

We next provide additional behavioral evidence for separate sites of learning within the cerebellar circuit. By probing the expression of learning as a function of pursuit speed across the trial-course of learning, we show that the linear pattern of generalization to pursuit speed we observed following a single learning trial (*Figure 4*) changes gradually but qualitatively across 1000 trials. We suggest that the site and/or mechanism of learning changes across a long trial-course.

We again deployed the dual-trial paradigm, but with the modification that the instruction trial in each pair of trials repeated the same fixed parameters of pursuit (5 deg/s or 20 deg/s), instruction magnitude (30 deg/s), instruction direction, and pursuit direction for the entire experimental session. We studied generalization across the gradual accumulation of learning by varying target speed in the pursuit direction randomly in the probe trial to be 5, 10, 15 or 20 deg/s, while keeping the pursuit direction the same as in the learning trials.

An example experiment demonstrates how generalization changed over a long trial-course when pursuit speed was 5 deg/s in the learning trials and was always equal or faster in the probe trials.

Averages of the trajectory of the learned eye velocity as a function of time during the first 250 trials confirmed that the learned response scaled with pursuit speed in the initial stages of learning (*Figure 9A*, left), in agreement with the data for single-trial learning in *Figure 4*. However, at the end of the same learning session, the expression of learning no longer scaled with pursuit speed in the probe trials (*Figure 9A*, right). The change in generalization across the session could not be explained by any systematic differences in the eye speed in the pursuit direction during the first versus last 250 trials (*Figure 9B*, continuous versus dashed traces).

The change in the generalization of the expression of learning in the probe trials occurred gradually (*Figure 10A*). Across 2250 trials (1125 dual-trial stimuli) in Monkey RE, the expression of learning declined for probes with pursuit speeds of 20 deg/s and remained constant for probes with pursuit speeds of 5 deg/s. By 2250 trials, the learned response did not depend on the target speed in the probe, in agreement with the right graph of *Figure 9A*. The observations were similar in Monkey YO, although he did not work for as many trials as Monkey RE (note the shorter x-axis). These results suggest that something about the inputs that are subject to plasticity or the mechanism of plasticity at the site of single-trial, short-term, learning may differ from those to the site of long-term pursuit learning.

We observed a qualitatively different pattern of generalization across learning blocks when pursuit speed was 20 deg/s in the learning trials and was always equal or slower in the probe trials. Now, the expression of learning increased slightly for probe trials with a pursuit speed of 20 deg/s and decreased as a function of trial when the pursuit speed in the probe trial was 5 deg/s (*Figure 10B*). Together, the results of learning with pursuit target speeds in the learning trials of 5 and 20 deg/s show a general rule: learning generalizes approximately linearly for pursuit speed across all timescales of learning when the pursuit speed in the probe trial is slower than the learning trial; when pursuit speed in the probe trial is faster than that in the learning trial, linear generalization early in learning is replaced gradually with a learned response that appears largely independent of pursuit speed after long-term learning.

To verify the statistical veracity of the effects in *Figure 10A and B*, we performed a regression analysis on the trial-course of generalization of each experiment (lines in *Figure 10A and B*). For each experimental condition, we measured the rate of change of the learned response across trials (*Figure 10C*). Here, a negative slope indicated that the learned response for a particular probe pursuit speed decayed over the experiment (e.g. 20 deg/s probe trials when the pursuit speed of the learning trial was 5 deg/s). Positive slopes indicated that the learned response measured in the probe trials tended to increase across the experiment, as was the case when the pursuit speed in both the learning trial and the probe trial was 20 deg/s (*Figure 10B*, red curves). A two-way ANOVA confirmed the differential effect of pursuit speed in the learning trials on the generalization properties of the motor memory. We tested for main effects of probe pursuit speed (5, 10, 15 or 20 deg/s), pursuit speed in the learning trials (5 deg/s or 20 deg/s), and their interaction. For both monkeys, we observed a main effect of pursuit speed in the learning trials (RE: $F_{(1, 32)}=46.8$, $p<10^{-7}$; YO: $F_{(1, 20)}=31.6$, $p<10^{-4}$), no main effect of probe speed (RE: $F_{(3, 32)}=1.39$, $p=0.26$; YO: $F_{(3, 20)}=0.33$; $p=0.81$), and crucially a significant learning pursuit speed by probe pursuit speed interaction (RE: $F_{(3, 32)}=14.2$, $p<10^{-5}$, YO: $F_{(3, 20)}=9.0$, $p<10^{-3}$). Thus, the relationship between pursuit speeds of the target in the learning and probe trials significantly affected the shape of generalization across trials.

To further validate the interaction between pursuit speed in the learning and probe trials, we conducted additional experiments in Monkey RE that tested other pursuit speeds in the learning trial (not shown). Here, different experiments used one of either 5, 10 or 20 deg/s as the consistent pursuit speed in the learning trials and probed with pursuit speeds selected randomly from the same three speeds. After 1000 learning trials, the expression of learning generalized approximately linearly as a function of slower pursuit speeds in the probe trial compared to the learning trial, but not for probe speeds faster than used in the learning trial, in agreement with *Figure 10A and B*. Thus, the rules for generalization depend not on whether the probe speed is the same or different from the pursuit speed in the instruction trials, but rather on whether it is faster or slower. These preliminary experiments also served as a control to ensure that the change in generalization across many trials was not a result of the alternation of learning and probe trials, because probe trials with the other two pursuit speeds were interspersed randomly among the learning trials with a ratio of 10 learning trials to every probe.

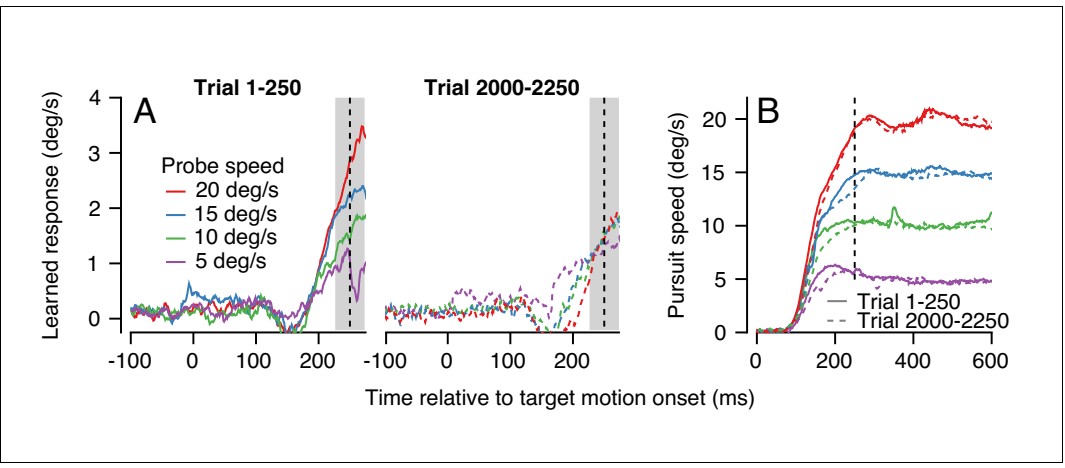

**Figure 9.** Changes in the pattern of behavioral generalization across an exemplar learning session in Monkey RE when pursuit speed in the learning trial is 5 deg/s. (**A**) Effect of pursuit speed in the probe trial on the expression of learning in the first 250 (solid, left) versus last 250 trials (dashed, right) of a long learning session. Data plot average learned eye velocity as a function of time from a single exemplar session in Monkey RE with pursuit speed of 5 deg/s and an instruction magnitude of 30 deg/s in the learning trials. Traces of different colors show the learned response for different pursuit target speeds in the probe trials. (**B**) Continuous and dashed traces show eye speed in the pursuit direction early versus late in the learning session during probe trials from (**A**). The online version of this article includes the following source data for figure 9:

**Source data 1.** Figure composer source data showing changes in the generalization to pursuit speed in the probe trial across a learning session.

## Functional differences in the fast versus slow sites/mechanisms of plasticity

In the previous section, we showed that learning generalizes differently in relation to pursuit target speed early versus late in a long learning block. Next, we analyze our data in a way that quantifies the shape of the pursuit speed generalization function at different stages of learning. We used the data from the long blocks of learning/probe pairs in *Figure 10*, where the pursuit and instruction speed were fixed in the learning trial but pursuit speed in each interleaving probe trial was selected randomly among 5, 10, 15, or 20 deg/s. Here, we computed the 'learning expression ratio' for each pair of trials as the magnitude of the learned response in the probe trial divided by the learning expressed in the preceding learning trial. The learning expression ratio is a single-trial metric of generalization, allowing us to probe the shape of generalization across long bouts of learning while controlling for the gradual accumulation of behavioral learning across the session. We then binned the resulting learning expression ratio by the ratio between the target pursuit speed in each probe trial and learning trial. When the probe trial and the preceding learning trial have the same pursuit speed (probe speed to pursuit speed ratio of 1.0), we would expect nearly complete generalization between the learning and probe trials (a learning expression ratio ~1.0). We refer to the relationship between the learning expression ratio and the ratio of the probe and learning trial pursuit speeds as the 'generalization function'.

In the first 250 trials of a long learning block (blue curve, *Figure 11A*), both monkeys demonstrated an approximately linear generalization function, paralleling our finding of single-trial generalization in *Figure 5*. Thus, early in learning, both monkeys expressed greater (less) learning in the probe trial than in the learning trial when pursuit speed in the probe trial was faster (slower) than in the preceding learning trial. As learning progressed through a long block of trials (*Figure 11A*, pink and purple traces), the shape of the generalization function changed gradually. After 1000 trials, probe trials with pursuit target speed faster than in the learning trial (probe speed to pursuit speed ratio >1.0) no longer showed enhancement of the expression of learning (learning expression ratio ~1.0). The changing shape of the generalization function was further accentuated in Monkey RE after 2000 trials when probe trials expressed less behavioral learning (learning expression ratio <1.0) if the pursuit speed was faster in the probe than in the learning trial.

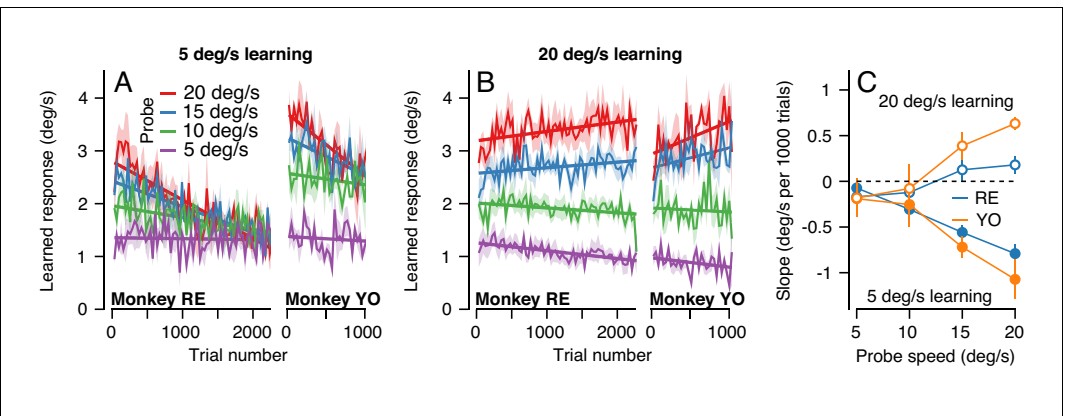

**Figure 10.** The pattern of generalization across a learning session depends on the relative speeds of target motion in the pursuit direction during learning versus probe trials. (A) Continuous changes in learned responses measured in probe trials across sessions when pursuit speed in the learning trial was 5 deg/s. Different colors show the responses for different target speeds in the probe trials. Note that the trial axis represents cumulative trials (learning and probe). (B) Generalization across learning sessions when pursuit speed in the learning trial was 20 deg/s. Different colors show expression of learning for probe trials with pursuit speeds of 5, 10, 15, or 20 deg/s as in (A). (C) Slopes of the trial-course of the expression of learning as a function of pursuit speed in the probe trial. Open and filled symbols show results for learning trials with pursuit speeds of 20 versus 5 deg/s. Instruction magnitude was always 30 deg/s. Error bars in (C) and error bands in (A) and (B) show ± SEM across days for each monkey.

The online version of this article includes the following source data for figure 10:

**Source data 1.** Figure composer source data showing the differential effects of pursuit speed in the learning trial on the generalization of learning.

That the generalization function changes across 2000 trials indicates either that the properties of a single site of learning change or that learning transfers between sites with different functional properties. Cognizant that factors such as non-linear synapses and recurrent inhibition can control the properties of the signals at a site of learning, the simplest way to model generalization is in terms of the signals conveyed by the afferent fibers whose synapses are subject to plasticity. *Figure 11* suggests that we can model learning under the assumption that the motor memory gradually transitions from an early site of learning where the inputs are linearly related to pursuit speed to a secondary site/synapse where the inputs are unimodally tuned for pursuit speed.

We completed our model of cerebellar learning (*Figure 12A*) by defining the functional properties of non-Purkinje cell inputs to FTNs. We simulated 50 inputs to the FTNs, each with a preferred direction either equal to or opposite the pursuit direction and a Gaussian tuned response for pursuit speed:

$$IN_n^i = \cos(\theta_n^e - \theta^i)\exp\left(-\frac{(\dot{E} - s^i)^2}{2\sigma^2}\right) \tag{9}$$

Here, for the $i^{th}$ input axon, $\theta^i$ is either 0 or 180 deg, $s^i$ is the preferred eye speed ranging from 0 to 50 deg/s, and $\sigma$ is the standard deviation of the Gaussian function. The eye movement vector on the $n^{th}$ trial is defined as $(\theta_n^e, \dot{E}_n)$. Note that any unimodal function (e.g. cosine) would yield comparable results to the Gaussian function.

The circuit-level learning system defined by the schematic in *Figure 12A* and *Equations 2-9* reproduces our data on both the trial-course of learning and the trial-course of the generalization of learning. The learned response, like the real data, shows a saturating relationship to instruction magnitude for single-trial learning (*Figure 12B*) and a linear relationship after 100 trials (*Figure 12C*). For learning with pursuit target motion at 5 deg/s, the expression of the learned response in the model generalizes differently early versus late in a 1000 trial learning block (*Figure 12D*). Early in the learning block, the learned response generalizes linearly to pursuit speeds regardless of the pursuit speed in the learning trials. Late in the learning block, the learned response

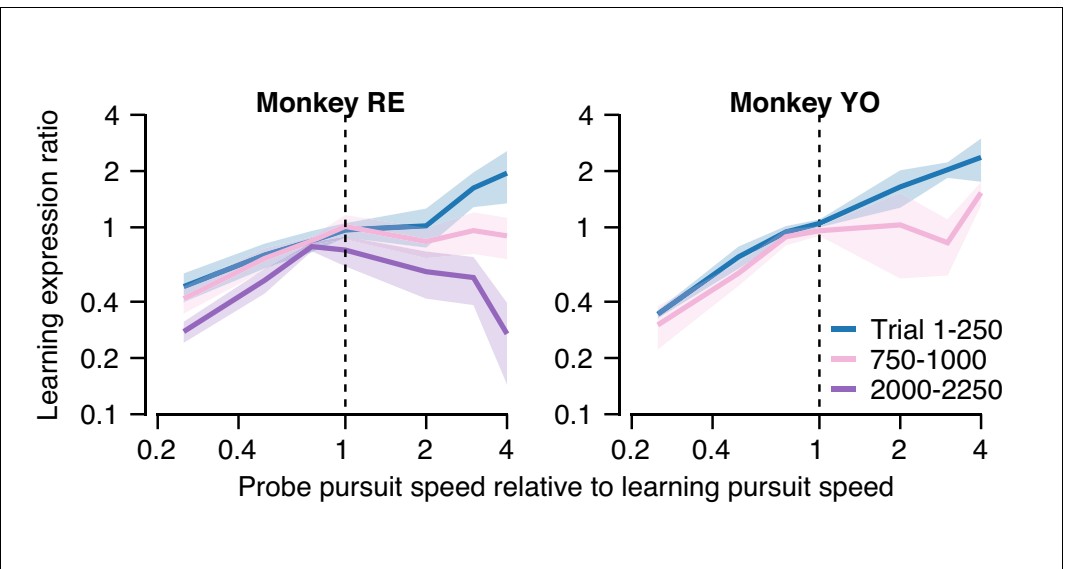

**Figure 11.** Quantification of changes in the generalization of learning expression across trials. Fraction of the behavioral learning that generalizes from a learning trial to the subsequent probe trial as a function of the ratio between the pursuit speed in the probe versus learning trial (both axes are shown on a log scale). Blue, pink, and purple curves show the shape of the generalization function early, in the middle, and late in long learning bouts, respectively. Error bands show ± SEM across sessions for each monkey.

The online version of this article includes the following source data for figure 11:

**Source data 1.** Figure composer source data characterizing changes in the shape of the generalization function across trials.

is largely identical across probe speeds. For learning with pursuit target motion at 20 deg/s, the learned response generalizes linearly across probe speeds lower than 20 deg/s for the entire duration of the learning block (*Figure 12E*). The model also makes predictions. It suggests that adaptive changes in the Purkinje cell responses should diminish over the course of thousands of trials (*Figure 12F,G*) as learning transfers to the FTN inputs (*Figure 12H,I*).

## Alternative models of cerebellar motor learning

We considered a number of other model forms besides our final cerebellar model (*Figure 12A*). The criteria for a plausible model were imposed by our data for single-trial learning (*Figures 3* and *4*) and repeated presentation of the same learning trial (*Figure 6*). Single-trial learning always showed a saturating relationship as a function of the error imposed by the instruction. Therefore, all candidate models were constrained by the fits in *Figure 3C*, ensuring that the measured learned behavior on trial 2 as a function of the error magnitude experienced on the first trial, $Y_2(e_1)$, was approximately equal to *Equation 1*. We then asked whether each candidate model could account for the nearly linear asymptotic learned response as a function of instruction magnitude after a block of 100 learning trials (*Figure 6C*).

We began by assuming the simplest model capable of reaching an asymptotic learned response that was less than the imposed instruction. This model featured a single learning process, $x$, with a retention parameter that allowed trial-to-trial forgetting:

$$\begin{aligned} x_{n+1} &= \alpha x_n + \Delta Y_n(e_n) \\ Y_n &= x_n \end{aligned}$$

(10)

Here, $\alpha$ is the retention factor where $\alpha=1$ indicates complete retention (no forgetting) and $Y_n$ is the measured behavioral response on trial $n$. The change in the measured amount of learning on each trial, $Y_n(e_n)$, is given by extending *Equation 1* from the case of single trial learning (trials 1 and 2) to trial-over-trial learning at any stage (trials n and n+1). The asymptotic response of this single learning process occurs when $Y_{n+1} \approx Y_n$, which is given by:

$$Y_\infty = \frac{Y_\infty(e_\infty)}{1-\alpha} \quad (11)$$

The form of the asymptotic response is a scaled version of the measured single-trial learning from *Figure 3C* and therefore cannot, by itself, reproduce the linear relationship between instruction magnitude and asymptotic learning that we measured after 100 consecutive learning trials. Any single-learning-process model that (1) has a static relationship between learning and error and (2) predicts saturating single-trial learning in *Figure 3* would fail to predict the linear asymptotic learning after 100 repeated learning trials in *Figure 6*.

Pursuit learning might align with other models of motor learning that also have posited two (*Lee and Schweighofer, 2009*; *Smith et al., 2006*) or more (*Kording et al., 2007*) learning processes. For our system, learning could be the net sum of two learning processes that run independently and in parallel: a fast-learning process, $x^f$, with rapid learning from error but limited retention and a slow-learning process, $x^s$, with strong retention:

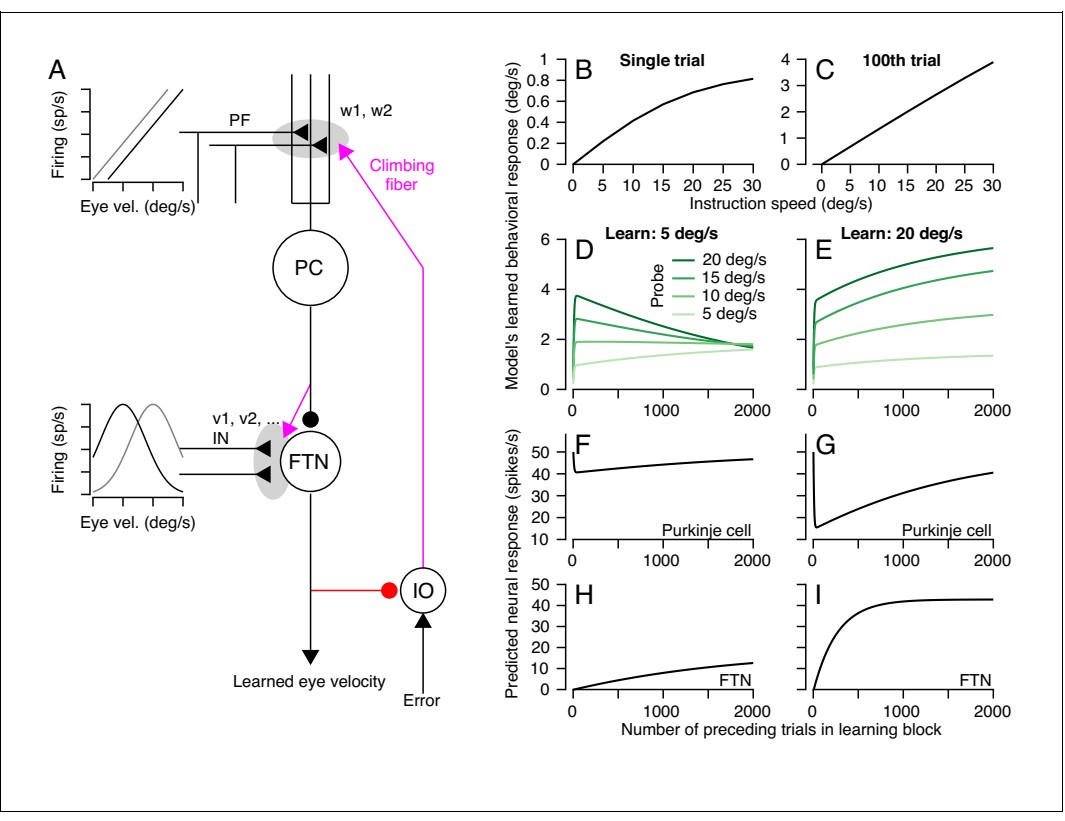

**Figure 12.** A circuit-level cerebellar learning model predicts many features of pursuit learning across timescales. (A) The schematic shows the structure of the model, which is described in *Equations 2-9* in the text. Magenta colors show pathways that transmit instructions for plasticity and the red pathway provides recurrent inhibition to the inferior olive. (B, C) Model's prediction for the size of learning as a function of instruction speed for single-trial learning (B) and after 100 learning trials (C). (D, E) Behavioral learning trial courses predicted by the model for learning sessions with a 30 deg/s instruction magnitude and pursuit speeds in the learning triasl of 5 deg/s (D) versus 20 deg/s (E). Different shades of green show predictions for the expression of learning using different pursuit speeds in probe trials. (F, G) Predicted learned Purkinje cell simple-spike responses across a learning session with pursuit speeds in the learning trials of 5 deg/s (F) and 20 deg/s (G). (H, I) Predicted size of inputs to FTNs after weighing through plastic changes in $v_i$, again across a learning session with pursuit speeds in the learning trials of 5 deg/s (H) and 20 deg/s (I).

The online version of this article includes the following source data for figure 12:

**Source data 1.** Figure composer source data showing predictions from a circuit-level model of motor learning.

$$\begin{aligned}
x_{n+1}^{f} &= \alpha^{f} x_{n}^{f} + \Delta Y_{n}(e_{n}) \\
x_{n+1}^{s} &= \alpha^{s} x_{n}^{s} + \eta^{s} e_{n} \\
Y_{n} &= x_{n}^{f} + x_{n}^{s}
\end{aligned} \tag{12}$$

A central requirement is that the retention of the slower-learning process is greater than the retention of the faster process, namely $\alpha^{s} \gg \alpha^{f}$, and the rate of learning is greater in the fast-learning process versus the slow-learning process, $Y_{n}(e_{n}) \gg \eta^{s} e_{n}$. As long as the fast-learning process shows a saturating relationship between error and single-trial learning, the model will reproduce the single-trial learning data in *Figure 3C*.

The two-independent process model in *Equation 12* fails to predict a linear relationship between the asymptotic response after extended learning and instruction magnitude. Given the retention values, we measured experimentally ($\alpha^{f} \approx 0.85$ and $\alpha^{s} > 0.95$), the value of $\eta^{s}$ must be very small to ensure that learning saturates at a level significantly below the imposed instruction after extended learning (*Figure 6A*). With this constraint ($\eta^{s} \approx 0$), we can derive the asymptotic response of the two-learning-process model from *Equation 12*:

$$Y_{\infty} = x_{\infty}^{f} + x_{\infty}^{s} = \frac{\Delta Y_{n}(e_{n})(1-\alpha^{s})}{(1-\alpha^{f})(1-\alpha^{s}+\eta)} + \frac{\eta I}{(1-\alpha^{f})(1-\alpha^{s}+\eta)} \approx \frac{\Delta x^{f}(e_{\infty})}{1-\alpha^{f}} \tag{13}$$

*Equation 13* represents a scaled version of the single-trial response shown in *Figure 3* and is largely identical to the asymptotic response predicted by a single-learning-process in *Equation 11*.

We expressly set out to model our data in a way that was driven by the known architecture of the cerebellar learning circuit. Adherence to the known circuit and the ability to reproduce a wide set of behavioral data with relatively simple assumptions are virtues of the model in *Figure 12A*. At the same time, we doubt that our successful model is unique and other strategies might (or might not) be able to reproduce our data as well. We give two examples. (1) A single-site model might work if the rate of learning is not constant but rather changes from trial-to-trial. It would be possible to account for some of our data in a single-state model where complex-spike-mediated depression of the parallel-fiber to Purkinje cell synapse intrinsically weakened or saturated across repeated identical learning trials. (2) Some models modulate learning based on Bayesian integration of noisy sensory feedback with internal estimates of the perturbation (i.e. the Kalman filter), experienced errors (*Herzfeld et al., 2014b*), or environmental consistency (*Gonzalez Castro et al., 2014*). Each of these models predicts that learning rate should increase for repeated presentations of the same learning stimulus versus single-trial learning, whereas we observed that the learning rate decreases over trials. However, one might contrive a model based on these principles where learning rate is lower when the stimulus regime is highly regular and consistent. In addition to their other failings, none of these approaches have the advantage of working seamlessly within the constraint of the architecture of the cerebellar learning circuit.

## Discussion

Using strategically designed behavioral experiments in rhesus monkeys, we have constrained a learning circuit model that implements four crucial principles of operation within the known neuroanatomy of the smooth pursuit eye movement system. The four principles of operation are: (1) early learning that occurs through a fast-learning process with poor retention, which likely employs climbing fiber mediated depression of the parallel fiber to Purkinje cell synapse; (2) over many trials, the motor memory transfers from the fast-learning process to a slow-learning process, with excellent retention, possibly in the deep cerebellar nucleus; (3) the inputs that are learned in fast- versus slow-learning processes have specific and different relationships between firing rate and eye velocity in the pursuit direction; and (4) recurrent feedback to the inferior olive from the output of the learning system modulates acquisition of learning via the fast-learning process in the cerebellar cortex. Even though our model is for cerebellar learning in pursuit eye movements, we suggest that the principles of operation are likely to generalize broadly, both to other cerebellum-dependent motor learning paradigms and to learning and memory systems elsewhere in the brain.

## Neural substrates of fast acquisition of learning

Our data add to the evidence for a fast-learning process with poor retention based on climbing-fiber mediated plasticity at the parallel to Purkinje cell synapse in the cerebellar cortex (*Medina and Lisberger, 2008*; *Yang and Lisberger, 2014*; *Nguyen-Vu et al., 2013*; *Kimpo et al., 2014*). Long-term depression at this particular synapse has been a staple of cerebellar learning for many years (*Ito, 2001*; *Marr, 1969*; *Albus, 1971*), and short-term forms of plasticity also occur at the same site (*Brenowitz and Regehr, 2005*; *Suvrathan et al., 2016*). Still, the tight link between the occurrence of a complex spike and neural learning in Purkinje cell simple-spike responses (and behavior) does not prove that the parallel fiber to Purkinje cell synapse is the mechanism of the fast-learning process and we need to remain open to the possibility that the actual site of plasticity may be upstream of the parallel fiber to Purkinje cell synapse.

We are able to measure the properties of the fast-learning process largely independent of other, slower, components of the recurrent learning circuit using the analysis of single-trial learning. Thus, we effectively measured the 'open-loop' features of the fast-learning process and discovered that the fast-learning process features a non-linear, saturating relationship between the magnitude of learning and the size of the movement error. We suspect that the non-linear relationship reflects the effects of the retinal slip on the probability and/or duration of the complex spikes that occur in Purkinje cells (*Mathy et al., 2009*; *Najafi and Medina, 2013*; *Yang and Lisberger, 2014*). However, we cannot exclude two alternatives. The asymptote of single-trial learning for the highest instruction magnitude could reflect the influence of inhibitory neurons on the learning signal (*Rowan et al., 2018*) and/or a simple limit on the maximum amount of plasticity that can be induced by a single climbing fiber input to most Purkinje cells. The saturating dependence of single-trial adaptation in our system on instruction magnitude or stimulus strength parallels findings of single-trial learning in other cerebellar-dependent adaptive behaviors (*Fine and Thoroughman, 2006*; *Herzfeld et al., 2014b*; *Marko et al., 2012*; *Hutter and Taylor, 2018*).

## Transfer of a motor memory between neural sites

Our finding of qualitatively different generalization functions early versus late in learning provides experimental support for two distinct sites of learning, one that learns quickly and predominates early versus a second process that learns slowly and therefore predominates late in a 1000-trial learning block. Evidence that learning transfers from the fast- to the slow-learning site, rather than proceeding independently in parallel, comes from our computational and theoretical analysis. The cerebellar circuit model with recurrent inhibition of the error input to the cerebellar cortex can explain linearization of the relationship between the size of learning and instruction magnitude after 100 learning trials, whereas a standard model with independent fast and slow-learning processes would require considerable modification to do so (see *Equations 12 and 13*). Thus, we think that the fast-learning process for pursuit learning occurs in the floccular complex of the cerebellum (*Medina and Lisberger, 2008*) and that the floccular target neurons in the vestibular nucleus (*Lisberger et al., 1994*) are likely the site of the slow-learning process and long-term storage. At this time, we cannot rule out the possible role of additional downstream areas.

There are multiple reasons to think that learning could transfer from the cerebellar cortex to the deep cerebellar nuclei, even though there is currently no direct neurophysiological evidence. Mechanisms of plasticity are abundant in the deep cerebellar nuclei and optogenetic modulation of the activity of Purkinje cells can adapt the vestibulo-ocular reflex, presumably through Purkinje-cell induced plasticity in the deep cerebellar nuclei (*Jang et al., 2020*; *Nguyen-Vu et al., 2013*). Recordings from Purkinje cells and deep cerebellar nucleus neurons following long-term vestibulo-ocular gain adaptation lead to the conclusion of neural learning in both sites (*Lisberger, 1994*) and are entirely compatible with the suggestion that the learned responses in Purkinje cells instruct learning in the deep cerebellar nucleus (*Miles and Lisberger, 1981*). For both eye movements (Pastor, *Pastor et al., 1994*) and classical conditioning of the eyelid response (*Garcia et al., 1999*), inactivation or lesions of the cerebellar cortex prevents adaptation but does not fully abolish learned responses from previous adaptation (*McCormick and Thompson, 1984*; *Perrett et al., 1993*), suggesting that the residual long-term learning resides outside of the cerebellar cortex. Finally, for learning in the optokinetic response, lesions of the flocculus abolish the learned response completely after short-term learning but only partially after long-term learning (*Shutoh et al., 2006*).

We think that the accessible neural substrate of pursuit learning affords the opportunity to study the neural mechanisms of memory transfer, a ubiquitous characteristic across memory systems. For example, different sites of early acquisition versus long-term storage exist for fear memories (*Rogan et al., 1997*), (*Do-Monte et al., 2015*), and in bird song (*Bottjer et al., 1984*; *Nordeen and Nordeen, 1993*; *Warren et al., 2011*). Indeed, the acquisition of a memory in one structure and the long-term storage of this memory in separate structures is clear from amnesic subjects, such as HM (*Zola-Morgan and Squire, 1990*), as well as from many more modern observations (*Tonegawa et al., 2018*).

## Differences in the characteristics of the signals that are subject to learning across trials

In our data, the early learned response generalizes linearly as a function of all pursuit speeds in the probe trials, even when the probe speed is faster than the pursuit speed in the learning trial. After 1000 trials of learning, the learned response scales only for probe speeds slower than pursuit speed in the learning trial. The literature contains other evidence of changes in the generalization of the expression of learning across timescales. In birdsong, learned changes in the pitch of one syllable generalizes differently early versus late in learning (*Warren et al., 2011*), suggesting that early learning in LMAN changes signals that are related to the overall song, while the late learning in the motor structure RA operates on signals related to the motor act of producing the syllable. In the vestibulo-ocular reflex, *Kimpo et al., 2005* showed different generalization for stimulus frequency for gain-up versus gain-down learning, suggesting different sites of learning based on modification of different vestibular signals.

After *Shadmehr, 2004*, we interpret the change in generalization of learning as possible evidence for a change in the properties of the neural signals that are subjected to learning. Our circuit model accounts for the change in generalization by assuming that a different set of eye speed signals is learned at Purkinje cells versus FTNs. The use in the model of different inputs to the sites of plasticity in the floccular complex and at the floccular target neurons suggests two features of the neural system that run counter to current doctrine. First, the general rule is that Purkinje cells and their targets in the deep cerebellar nuclei receive identical inputs from mossy fibers. In the floccular complex, however, this does not seem to be the case: brainstem areas that send afferent mossy fibers to the ventral paraflocculus region of the floccular complex send only sparse projections to the location of FTNs in the vestibular nucleus (*Osanai et al., 1999*; *Nagao et al., 1997*). Second, the general rule in the oculomotor system is the firing rate is linearly related to eye position and velocity, and this is true for mossy fibers in the floccular complex (*Miles et al., 1980*; *Lisberger and Fuchs, 1978*). Thus, there is no obvious precedent for neural responses that are unimodally tuned for eye velocity. Limited recordings from FTNs during pursuit learning *Joshua et al., 2013* demonstrate a diversity of FTN velocity-dependent responses to a 30 deg/s pursuit stimulus, perhaps consistent with a distribution of preferred speeds across the neural population. We realize that the assumptions in the model may not reflect the actual implementation of the system in the brain, and that the differences in generalization at different sites could reflect properties of synapses or plasticity mechanisms rather than of the input signals that are subject to plasticity.

## Recurrent inhibition of the fast-learning process learning signal

We have always been troubled by the fact that pursuit learning, like other forms of motor learning (*Krakauer et al., 2000*; *Tseng et al., 2007*; *Vaswani et al., 2015*), never reaches an amplitude that completely eliminates the error created by the imposed instruction. Even after more than 1000 learning trials, the adapted pursuit eye movements shows residual errors than are in excess of 50% of the imposed instruction (*Hall et al., 2018*).

We suggest that motor learning never fully adjusts for the instruction because of active inhibition of the instruction for learning *Kenyon et al., 1998*). Via the known double-inhibitory pathway from Purkinje cells to the inferior olive via the deep cerebellar nucleus (*Houck and Person, 2014*; *Najac and Raman, 2015*; *de Zeeuw et al., 1988*), learned depression of Purkinje cell simple-spike firing could reduce the effectiveness of a given movement error by attenuating the probability and/or duration of climbing fiber inputs to the cerebellar cortex. Previous neurophysiological observations over 100 trials of pursuit learning demonstrate a reduction in the probability of complex spikes,

hinting at possible inhibition of the olive (*Medina and Lisberger, 2008*; *Yang and Lisberger, 2017*). Our data and modeling render unlikely the alternate explanation that incomplete adaptation is the result of a competition between the learning signals and a restoring force due to forgetting (*Smith et al., 2006*; *Albert et al., 2019*). During error-clamp trials, we measured the retention parameter after 1000 trials to be close to 1.0, corresponding to almost complete retention.

We imagine that recurrent inhibition of the error inputs to the fast-learning process promotes slow, conservative learning that does not respond hastily to noisy inputs. We do not understand fully why it is advantageous for the fast-learning process to remain inhibited after learning has been transferred from the cerebellar cortex to FTNs. Perhaps the goal is to force other parts of the pursuit circuit to take over some of the learning under conditions where permanent changes seem desirable, by transferring some of the learning away even from the FTNs. Alternatively, our somewhat rarified stimulus conditions may not be engaging the neural learning circuit fully so that we are seeing only a part of the full learning capability of the system.

### 'Principles of operation' versus 'sites and mechanisms of plasticity'

The cerebellar learning circuit is stereotyped, well-known, accessible for analyses at the level of the activity of single neurons, and engaged in quantitatively-defined learning behavior. We see it as an exemplar where we have the greatest potential to determine not just the sites and mechanisms of plasticity, but also how the essential neural circuit exploits the local mechanisms of plasticity and converts them into the global deliverable of an adaptive change in behavior. Similar analysis and thinking are going to be essential in all other learning systems and neural circuits before we can say that we 'understand' any form of learning and memory. The particular behavioral system we study affords the huge advantage that we can constrain models by both the architecture of the essential circuit and data on the operation of the system. Our data and computational thinking assemble a statement of the principles of operation of this learning neural circuit. Once additional behavioral and neurophysiological evidence further refine these principles of operation, we will have a strong statement of how one learning circuit works.

## Materials and methods

Two adult *macaca mulatta* monkeys (12–16 kg, both male) served as the subjects for all experiments. All experimental procedures were performed in accordance with the *Guide for the Care and Use of Laboratory Animals* (1997) and had been approved in advance by the *Institutional Animal Care and Use Committee* at Duke University (Protocol A085-18-04). Prior to the experimental procedures, monkeys were deeply anesthetized with isoflurane and a head-holder was implanted using sterile technique to prevent head motion during experimental sessions. In a second sterile surgical procedure, a coil was sutured to the sclera of one eye (*Ramachandran and Lisberger, 2005*), allowing the high precision recording of eye kinematics using the scleral coil technique (*Robinson and Fuchs, 1969*). Monkeys received analgesics for several days following each surgical procedure. Following recovery, monkeys were trained to pursue a moving target in exchange for liquid reward. Both monkeys had substantial experience performing smooth pursuit eye movements before data collection.

### General experimental procedures

Monkeys were positioned with their heads fixed 30 cm in front of a CRT monitor (resolution: 2304 × 1440, framerate: 80 Hz). The monitor subtended 58 × 46° of the monkey's visual field. All experiments took place while the monkey was in a dimly lit room. The visual target for all experiments was a 0.5° diameter black dot presented on a light grey background (approximate luminance 32.9 cd/m$^2$). The motion of the target at each frame was controlled by our laboratory's custom 'Maestro' software.

Separate voltage signals from the scleral coil system, corresponding to the monkey's horizontal and vertical eye position, were digitized at 1 kHz. We also digitized eye velocity signals obtained by passing the eye position voltages through an analog differentiator circuit with a low-pass filter that reduced the amplitude of signals above 25 Hz (−20 dB/decade). All signals were stored for later off-line processing and analysis.

## Behavioral procedures

At the start of each trial, the monkey fixated within ±3° of a target at the center of the screen for a random interval, chosen from a uniform distribution of 400–800 ms. After the fixation period, the target instantaneously jumped eccentrically by 0.15 $v_t$ degrees, and then began moving at a constant speed, $v_t$, in the opposite direction, termed the 'pursuit direction' (step-ramp paradigm of *Rashbass, 1961*). The eccentric target jump minimizes the number of catch-up saccades during the initial tracking of the target. During baseline, probe, and washout trials, the target proceeded at the constant 'pursuit speed' for 850 ms before stopping. The monkey then fixated the target at its final position for an additional 200 ms. In exchange for appropriate tracking of the target as well as fixating the target within ±3° at the end of the trial, the monkey received a small fluid reward. If the monkey broke fixation during the initial fixation period or failed to adequately track the moving target and fixate its final position, the trial was aborted immediately and the monkey received no reward. Aborted trials were not included in the data analysis.

To induce cerebellar-dependent motor learning, we used a direction learning paradigm (*Medina et al., 2005*). In 'learning trials' (*Figure 2A*, dotted line, *Figure 2B,D*), the target proceeded in the original pursuit direction for 250 ms, before the addition of an orthogonal velocity component, termed the 'instruction.' We call the direction of the orthogonal instruction the 'learning direction,' since the instruction induces visual motion that serves as an error and drives cerebellar-dependent motor learning. During the instruction, we suspended the eye position requirements for liquid reward. In some experiments, we varied the speed of the instruction. In all experiments, the instruction had a duration of 400 ms, so that it was followed by 200 ms of target motion in the original pursuit direction and then 200 ms of fixation. Reward was not contingent upon generation of a learned response. Monkeys typically worked for 1500 to 3000 successful trials per experimental session.

To measure the amount of trial-to-trial forgetting in the absence of error, we used 'error-clamp' trials. During error-clamp trials, the target proceeded in the original pursuit direction, as in a probe trial. However, the location of the target in the direction orthogonal to the pursuit direction (the learning direction) was 'clamped' to the animal's eye position, such that the position of the target on each of the monitor's frames matched the animal's current eye position in the learning direction (*Figure 3D*). Target stabilization in the learning direction removes any errors that could drive unlearning and allowed us to measure the intrinsic retention/decay of motor memories without any stimulus for learning.

## Dual-trial paradigms for investigating short-term learning and generalization

To separately investigate the characteristics of the signals that drive learning of a motor memory from those that affect generalization on the timescale of a single-trial, we developed a 'dual-trial' experimental paradigm as a variant of our previously described paradigm for single-trial learning (*Yang and Lisberger, 2010*). Our modified paradigm uses pairs of trials, where the first trial, *n*, is the learning trial that provides a direction-change instruction (*Figure 2A*, dotted line, *Figure 2B*) and the second trial, *n+1*, is a probe trial without an instruction (*Figure 2A*, solid line, *Figure 2C*). Both the learning and probe trial in each dual-trial pair used the same pursuit direction, chosen randomly for each pair of trials from the cardinal axes (0°, 90°, 180°, or 270°). For some experiments, the speed of the target in the pursuit axis (i.e., the pursuit speed) was identical in both the learning and probe trials. In other experiments, we strategically altered the pursuit speed in either the learning trial or the probe trial. The direction of the instruction in the learning trial was chosen randomly to be either +90° or −90° relative to the pursuit direction. Due to the random directions of the initial pursuit and the instruction within a daily experiment, learning did not accumulate across trials. We explicitly prohibited the consecutive occurrence of pairs of trials with the same pursuit and instruction directions. To quantify the behavioral learning expressed in the second trial of each pair, we measured the 'learned response' from the eye velocity in the direction of the instruction in the preceding learning trial (i.e., the learning direction). Monkeys typically performed 750 to 1500 pairs of trials in a given experimental session.

We extended our dual-trial paradigm to investigate changes in the generalization of learning to pursuit speed across long blocks of trials. Now, we allowed the learning to accumulate by using a

consistent pursuit direction in both the learning and probe trials across the entire experimental session. The pursuit direction for both the learning and probe trial was chosen randomly from the cardinal axes (0˚, 90˚, 180˚, or 270˚) for each experimental session. Within an experimental session, the pursuit speed during the learning trial was either 5˚/s or 20˚/s. To measure the generalization of learning to pursuit speed, the speed of each probe trial was chosen randomly (5, 10, 15, or 20 deg/s). For all experiments that used the dual-trial paradigm, we report statistics for each monkey across days/experimental sessions.

### Repeated-instruction paradigms

To measure changes in the acquisition of a motor memory across timescales longer than a single-trial, we used blocks of trials with consistent instruction statistics. For each 'learning block,' we chose a single direction of pursuit for all trials from one of the cardinal axes. Monkeys performed a series of baseline trials without an instruction, followed by a series of learning trials with an imposed instruction in a consistent learning direction that was orthogonal to the pursuit direction. The behavioral learning was subsequently extinguished by presenting washout trials that did not feature an instruction. The number of washout trials was always equal to or larger than the number of the preceding learning trials. Because the direction of the instruction was consistent across the learning trials, we define the behavioral learning on each trial as the speed of the eye movement in the direction of the instruction experienced during the learning trials. After a complete learning block, including washout, a new pursuit direction and instruction direction were chosen randomly for the subsequent block. Monkeys typically performed multiple repetitions of repeated-instruction learning paradigms in a given experimental session. No two consecutive learning blocks were allowed to have the same pursuit and instruction directions. For all experiments in a repeated-instruction design, we report statistics for each monkey across repetitions of the learning paradigm.

### Data analysis

To ensure that saccades did not contaminate our estimate of the learned pursuit response, we identified and removed saccades from eye velocity traces. We identified saccades using a combined speed and acceleration threshold: any instance when the speed of the eye exceeded 20 deg/s and the eye acceleration exceeded 1250 deg/s/s was labeled as a saccade. Time points were marked as a saccade from 10 ms before the first time point that exceeded joint velocity and acceleration thresholds to 10 ms after the final time point that exceeded both thresholds. We eliminated these intervals from data analysis by treating them as missing data. Because the target motions were designed to minimize their occurrence, saccades were typically infrequent in the analysis intervals and occurred at variable times.

To summarize the magnitude of behavioral learning on a single-trial, we averaged the speed of the eye in the learning direction from 25 ms before to 25 ms after the time of the instruction, 225–275 ms after the onset of pursuit-direction target motion (grey shading in *Figure 2B,C*). Even during trials that included an instruction (e.g. during block-wise learning), this measurement provides an estimate of the learned response because the effects of visual feedback and online corrections in pursuit movements do not appear until at least 75 ms after the onset of the instruction. This measurement of motor learning is consistent with previous pursuit direction learning experiments (*Hall et al., 2018*; *Yang and Lisberger, 2010*; *Yang and Lisberger, 2017*).

### Statistical analysis

We used two tailed t-tests to compare sample means for conditions that were not sampled within the same experimental session, with the significance level set at $p=0.05$. Remaining tests with conditions that were simultaneously sampled within an experimental session were performed using a repeated measures analysis of variance (RM-ANOVA).

## Acknowledgements

The work was supported by NIH grants R01-NS092623 (SGL), K99-EY030528 (DJH), and F32-NS103216 (NJH). We thank Stefanie Tokiyama and Bonnie Bowell for animal assistance and Steven Happel for IT support.

## Additional information

### Funding

| Funder | Grant reference number | Author |
|---|---|---|
| National Institute of Neurological Disorders and Stroke | R01NS092623 | Stephen G Lisberger |
| National Institute of Neurological Disorders and Stroke | F32NS103216 | Nathan J Hall |
| National Eye Institute | K99-EY030528 | David J Herzfeld |

The funders had no role in study design, data collection and interpretation, or the decision to submit the work for publication.

### Author contributions

David J Herzfeld, Conceptualization, Data curation, Software, Formal analysis, Investigation, Visualization, Methodology, Writing - original draft, Writing - review and editing; Nathan J Hall, Conceptualization, Funding acquisition, Investigation, Methodology, Writing - original draft, Writing - review and editing; Marios Tringides, Conceptualization, Investigation, Methodology, Writing - review and editing; Stephen G Lisberger, Conceptualization, Resources, Supervision, Funding acquisition, Investigation, Methodology, Writing - original draft, Project administration, Writing - review and editing

### Author ORCIDs

David J Herzfeld https://orcid.org/0000-0001-9910-0658
Nathan J Hall https://orcid.org/0000-0003-3690-7395
Stephen G Lisberger https://orcid.org/0000-0001-7859-4361

### Ethics

Animal experimentation: All experimental procedures were performed in accordance with the Guide for the Care and Use of Laboratory Animals (1997) and had been approved in advance by the Institutional Animal Care and Use Committee at Duke University (Protocol A085-18-04).

### Decision letter and Author response

Decision letter https://doi.org/10.7554/eLife.55217.sa1
Author response https://doi.org/10.7554/eLife.55217.sa2

## Additional files

### Supplementary files

• Source code 1. Jupyter notebook implementing a single-process-model of pursuit learning. Implementation of the model shown schematically in *Figure 5A*, corresponding to a single-learning-process model with a single site of plasticity at the parallel fiber to Purkinje cell synapse. The source code is released under the MIT software license.

• Source code 2. Jupyter notebook implementing a circuit model of cerebellar pursuit learning. Implementation of the model shown schematically in *Figure 8B*, featuring two sites of learning in the cerebellar circuit. The source code is released under the MIT software license.

• Transparent reporting form

### Data availability

The data for each figure is included in a Figure Composer FYP file and can be viewed, exported, and further analyzed using the freely available Figure Composer tool (https://sites.google.com/a/srsci-comp.com/datanav/figure-composer). This tool is platform agnostic and runs on Windows, Mac, and Linux systems. The source code used to generate the cerebellar model results (Figures 5 and 8) is

included as a Jupyter notebook. This source code makes use of Julia but can be viewed without installing Julia.

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
