## [Decision Letter]

**Acceptance summary:**

Using a series of behavioral experiments and modeling analyses, the authors suggest that cerebellum-dependent motor skill learning is supported by separable "fast" and "slow" learning processes. The manuscript systematically and tightly integrates four functional principles governing this process, consistent with previous experimental evidence and models of the cerebellar anatomy and physiology. Incisive behavioral studies like this one, which provide clear insights/constraints on neural mechanisms, have become all too rare.

**Decision letter after peer review:**

Thank you for submitting your article "Principles of operation of a learning neural circuit" for consideration by *eLife*. Your article has been reviewed by three peer reviewers, and the evaluation has been overseen by a Reviewing Editor and Joshua Gold as the Senior Editor. The following individual involved in review of your submission has agreed to reveal their identity: Samuel McDougle (Reviewer #1).

The reviewers have discussed the reviews with one another and the Reviewing Editor has drafted this decision to help you prepare a revised submission.

Summary:

In this paper, Herzfeld and colleagues take a computational and psychophysical approach to investigate mechanisms of smooth-pursuit learning behavior in monkeys. Using a series of behavioral experiments and modeling analyses, the authors suggest that learned responses appear to be modulated by the learning environment in a manner that reflects two underlying, separable "fast" and "slow" learning processes, and transfer from the "fast" to the "slow" system. First, the authors show that the relationship between the adaptive response and the size of the experienced error is modulated by the consistency of the environment; that is, the scaling function saturates when the error changes magnitude every trial, but becomes linear if particular errors are experienced repeatedly. These effects are best captured by their dual-process model rather than a single process with a fixed learning algorithm. Next, in a series of speed-based generalization experiments, the authors show that a) generalization functions appear to change with training, suggesting a shift in the relevant inputs to the learning circuit, and b) there are differential changes in generalization patterns over time based on the particular speed of the training stimuli. Finally, the authors formalize their theory in a cerebellum-inspired circuit model of pursuit learning to generate precise predictions about expected neurophysiological instantiations of their behavioral observations.

The manuscript contains a very impressive set of careful, well-designed experiments, and a thought-provoking modeling analysis. The main question is compelling and the reviewers were, for the most part, convinced that the conclusions were largely supported by the data. However, the reviewers identified several points that should be addressed more carefully. Moreover, in its current form, the reviewers found the manuscript difficult to read, even for researchers with considerable relevant modeling and experimental experience. As currently written, it has several logical leaps, making it difficult to follow. It would benefit from restructuring to better guide the reader through the results and justify the modeling.

Essential revisions:

1) The manuscript should be thoroughly edited to improve clarity of the manuscript. As currently written, it has several logical leaps and is quite difficult to follow. The difficulty may arise in part from switching between model and experiment, and an uncertainty about where the results are going. At various points new bells and whistles are added to the model, and it is not always clear why these particular parameterizations are used. Two suggestions for reorganization, which could help with the clarity of the narrative:

– Present the circuit model at the beginning, and then proceed to constrain and test the model. This may be the better option, because dual-site plasticity is already an existing hypothesis for cerebellar function.

or

– Present all the experimental data first, and then build the model.

Additional, more detailed comments about where clarity could be improved are provided below.

2) An expanded, formal model comparison would better in the main text versus the Materials and methods section.

a) Provide a more clear explanation of why an independent fast and slow system can't work.

b) The "dual-process" claim would be strengthened by testing alternative models beyond just the single process model and the non-interaction dual-process model. Is a dual-process model more parsimonious than alternatives that, say, posit a single process with dynamic learning rates (or retention factors) that change systematically over time, as a function of environmental consistency (Castro et al., 2014), or both? Indeed, such models are commonplace in other domains, like reinforcement learning (Iigaya et al., 2018; Behrens et al., 2007; Nassar et al., 2010). Alternative dynamic models (e.g., Kalman filter) might perform significantly worse than the author's dual-process model in explaining the data, but this is an empirical question.

3) Additional explanation is needed regarding the generalization. The learning model does not appear to explain the observed generalization effects (20 vs. 5 deg/s differences, and the gradual shift from linear to gaussian). Although this is addressed in the circuit model's predictions, it would be useful to find a way to validate either the descriptive or circuit model via simulations of the generalization behavior. In addition, the presentation should be improved to more clearly guide the reader through the logic of this section.

a) The first main model conclusion, which is that a slow learning system is required to explain the change in speed generalization to a linear relationship, is not explained well. Intuition is not provided for why the slow system produces a linear relationship when taught by the fast system which does not. When this modification to the model is introduced, multiple features are added (a second system, and feedback control of the error signal), and it isn't clear which of these, or both, are necessary to obtain linearity. Starting in the subsection “Strategy for testing the conceptual model of motor learning”, the authors explain the strategy saying "inputs", which are not illustrated in their conceptual model (Figure 5A). Readers, especially those who are not familiar with the cerebellar circuit, are left to wonder, because this strategy needs imagination based on a real circuit which has an eye velocity related-firing of input, plasticity site, input and plasticity-dependent output for expression of learned behavior. Readers may understand at last when they see Figure 10A, which is the cerebellar learning circuit.

b) The description of the inference of input tuning when describing learning generalization is a bit obtuse. "The changing shape of the generalization function was further accentuated in Monkey RE after 2000 trials when probe trials expressed less behavioral learning (learning expression ratio <1.0) when the pursuit speed was faster in the probe than in the learning trial, suggesting Gaussian generalization." "Gaussian generalization" not defined anywhere. It should be, and the particular form used should be motivated. I assume what the authors mean is that it supports some kind of unimodal tuning of the input. Why can't the form be directly inferred from data, rather than assumed to be Gaussian? In general, the fitting to Equation 6 seems rather arbitrary.

c) Regarding the generalization of learning to different probe speeds: There should be more discussion of this result from an algorithmic perspective and possible/alternative explanations. Can the results be explained by "confidence" about the speed during the learning trial and its relationship to the probe speed that increases over learning and, at the end of learning, results in the learning expression only generalizing to the same speed ("Gaussian" generalization)? Such a discussion could make more intuitive what is otherwise a difficult and technical section that leads to strange conclusion about the model (see next comment).

d) One of the hardest to swallow aspects of the model is the necessity of extremely different speed tuning of the inputs to the fast and slow systems. It is especially hard to swallow because these inputs both ultimately arise presumably from mossy fibers that contact nuclear cells directly or contact granule cells. In such a model, it would seem that either different mossy fibers for each system would be required, or some complicated transformation would need to happen at the granule cell. The authors should expand their discussion of how this might come about, and discuss whether there are other ways that the results might be explained without this strong assumption.

4) Results should be discussed in the context of previous work from other learning and memory systems. This paper highlighted four principles that explain learning of smooth eye responses that predict a change in target direction. Previous work on these principles in other learning and memory systems should be discussed.

5) Several figures show data only from one monkey. Please justify this.

---

## [Author Response]

Essential revisions:1) The manuscript should be thoroughly edited to improve clarity of the manuscript. As currently written, it has several logical leaps and is quite difficult to follow. The difficulty may arise in part from switching between model and experiment, and an uncertainty about where the results are going. At various points new bells and whistles are added to the model, and it is not always clear why these particular parameterizations are used. Two suggestions for reorganization, which could help with the clarity of the narrative:– Present the circuit model at the beginning, and then proceed to constrain and test the model. This may be the better option, because dual-site plasticity is already an existing hypothesis for cerebellar function.or– Present all the experimental data first, and then build the model.Additional, more detailed comments about where clarity could be improved are provided below.

We have followed the reviewers’ first suggestion. We now begin the Results section by describing the cerebellar circuit essential for pursuit as well as summarizing previous neurophysiological datasets that are the foundation for our ultimate model of learning. We have completely removed the original conceptual model in favor of introducing a complete circuit model that we constrain a step at a time via the behavioral data as we build the full model to account for the full dataset. The benefit of this approach is that (1) each experiment is specifically linked back to the hypothesized neurophysiological substrate and (2) it now is clearer what features of the data link to each specific component of the model.

2) An expanded, formal model comparison would better in the main text versus the Materials and methods section.

We have moved the model comparison out of the Materials and methods section and instead to the end of the Results section. We also have extended this section to provide additional clarity about why a standard two state model is unable to account for our data. We also now include a paragraph comparing our resulting model with more complex “single-state” models of learning. While the revised manuscript does not provide a formal comparison between our model and others (i.e., comparing AIC values), we hope it is sufficient to provide intuition for the reader for why these models would fail to adequately explain our data. We also explain more clearly that our model satisfies an additional constraint that the other models lack – it is true to the known organization of the cerebellar learning circuit.

a) Provide a more clear explanation of why an independent fast and slow system can't work.

The short answer to this concern is that we cannot conclusively refute the idea of independent fast and slow learning processes using the data we have collected. Indeed, we share the reviewers’ intuition that there are other models that can account for our data, and one of them might have independent fast and slow learning processes with parameters that adapt over the course of a long block of learning trials. Our goal in this paper was not to explore fully all possible models that could explain a complex dataset, but rather to frame the success of fairly simple principles placed within the known architecture of the cerebellar learning circuit. We have tried to write more clearly so as to set expectations appropriately and to be clear in the Discussion about what we have and have not demonstrated.

There also is a longer answer and we have tried to touch on its elements in the revised version of the paper. We now provide a more detailed explanation of why the standard two-state model posited by Smith, Ghazizadeh, and Shadmehr, 2006, which features independent fast and slow processes, cannot account for our data. Even with some modifications, that model cannot account for both the saturating relationship between single trial learning and error (Figure 3) and the linear relationship between asymptotic behavior and error after 100 trials (Figure 7B). As mentioned above, it would be possible to contrive an independent 2-state model if we allowed the parameters of the model to change across a long block of learning trials, or by assuming complicated but implausible relationships between learning and error for the slow process. But, the point of our paper is not an extensive examination of all possible models that could fit our data, but rather an explanation of how a model based on the known architecture of the cerebellar learning circuit could work.

We have added Figure 8C to allow us to explain intuitively another failing of many models with either a single learning process or 2 independent learning states. Without some form of inhibition, these models predict that the asymptotic learned eye velocity should be much closer in amplitude to the imposed error. When we add active inhibition of the fast learning process through known inhibitory feedback from the cerebellum to the inferior olive, this feature of the data emerges gracefully. We have covered this point explicitly in the Results section.

Finally, as is now pointed out more clearly in the paper, the neuroanatomy of the essential cerebellar pursuit circuit and considerable previous research show how learning could be transferred from Purkinje cells to FTNs within the known architecture of the circuit. Therefore, we view a “transfer” of learning from the Purkinje cells to the vestibular nucleus as the most parsimonious explanation for our data, and also as an explanation that makes testable predictions.

b) The "dual-process" claim would be strengthened by testing alternative models beyond just the single process model and the non-interaction dual-process model. Is a dual-process model more parsimonious than alternatives that, say, posit a single process with dynamic learning rates (or retention factors) that change systematically over time, as a function of environmental consistency (Castro et al., 2014), or both? Indeed, such models are commonplace in other domains, like reinforcement learning (Iigaya et al., 2018; Behrens et al., 2007; Nassar et al., 2010). Alternative dynamic models (e.g., Kalman filter) might perform significantly worse than the author's dual-process model in explaining the data, but this is an empirical question.

Again, there is both a short and a long answer to this concern. First, the short answer: Our goal in the present paper was to identify the general properties of the underlying learning system using both behavioral data and the architecture of the cerebellar learning circuit as constraints. Therefore, we chose not to explore comparisons of model fits for a wide range of models that are not constrained by the known circuits. Instead, we have focused on how to account for the data with a model based on the known cerebellar circuit. In our revision, we have been more explicit about the virtues of the circuit-based modeling approach, while also touching on some of the alternatives mentioned here in a new paragraph that is now the last paragraph of the Results section.

The long answer involves comparisons with specific classes of models in a level of detail that we feel would not be appropriate for the paper. Indeed, the description given below takes ~800 words and, to us, does not seem appropriate for the Results or Discussion of our paper.

*Models that incorporate environmental consistency:* The reviewers suggest a comparison of our dual-process model of cerebellar adaptation with the results of Gonzalez Castro et al., 2014. The authors of this study suggest that increasing environmental consistency drives increases in learning from error. We can reasonably assume from Figure 3D of their publication that the relationship between environmental consistency and learning rate is likely linear so that the single trial learning rule is: xn+1=αxn+ηnen where represents the error experienced on trial *n* (i.e., the difference between the predicted motor output on trial *n* and the observed motor output) and is a linear scaling factor that adjusted the magnitude of the trial-to-trial learning based on the lag-1 auto-correlation of the preceding trials. This model fails qualitatively to account for our data. For our single trial learning experiments, the lag-1 auto-correlation of the perturbation is near zero due to the randomization of the task. In contrast, for 100 repetitions of the same learning trial, the lag-1 auto-correlation begins close to zero and then gradually increases as the perturbation becomes increasingly consistent. Thus, Castro et al. would predict a progressive increase in the amount of trial to trial learning across the 100-trial learning block whereas this is not the case in our system (Figure 8C). Our simulations fixed the error-sensitivity term, , such that its value corresponded to those measured during single-trial learning (lag-1 auto-correlation of approximately zero). Allowing the error-sensitivity parameter to increase with increasing lag-1 auto-correlations would only exacerbate the over-prediction of the learned response after blocks of trials. We also note that the same issue holds for alternative models that manipulate error-sensitivity based on the statistics of the perturbation and/or error (Herzfeld et al., 2014).

The model proposed by Nassar et al., 2010, specifically addressed changes in the updating of reward predictions based on reward prediction errors. While our reward landscape was not dependent on the adapted state of the monkey (rewards were never contingent on the animal’s generation of a learned response), one might suggest that the same model might apply with sensory prediction errors as substitutes for reward prediction errors. One of the primary features of the reduced model in the Nassar paper was the detection of change-points in the structure of the task, something that did not occur in our task for either sensory or reward prediction errors. Learning experiments with many repetitions of the same learning trial were performed in separate experimental sessions than single-trial learning experiments with randomized learning parameters. There was never an experimental session in which the experimental paradigm changed from random to non-random (or the reverse).

Single-process Kalman filter/Bayesian integration: An ideal way to modulate the learning rate of a single process model might be to use the Kalman gain as the learning rate. Here, ηn∝σx/[σx+σy], where σx is the estimate of the state noise and σy is the noise in the observation. In this formulation, when the variance of sensory evidence is large (σy), the Kalman gain is small, relying more on the previous estimate of learning state, *x.* In contrast, when the variance associated with the state estimate is larger (σx), the Kalman gain should be larger, resulting in significantly more updating of the state estimate with sensory evidence. In our experiments we did not specifically manipulate sensory evidence, but it might be possible to think of the statistics of the direction of the motor error as homologous to the reliability of the sensory error. Then, we would again predict lower learning rates for single-trial learning than for blocks that repeat the same learning trial 100 times, again contradicting our data. Even if increased learning creates larger amounts of signal-dependent state noise, the effect would be to increase the learning rate for longer bouts of repeated learning, again contradicting our data. The same would be true of any single-process learning model that derives the learning rate from Bayesian statistics. For instance, the δ-learning rule for reinforcement learning proposed by Behrens et al., 2007, relies on the updating of the value of two targets via a reward prediction error signal. Modifying this model to update the belief about the magnitude of the instruction via sensory prediction error results in the same predictions as the Kalman filter, described above.

Other multi-process models of learning: Iigaya et al., 2019, propose a model of reward learning using multiple timescales as a method to explain deviations from matching behavior during foraging tasks. In their conceptualization, there are two (or more) learning processes for each rewarding target. The time constants associated with these learning processes differ, updating the local income for each target at different timescales. For each target, the estimated income is then the weighted output of each of these processes. The weights for each timescale and target where then fitted using maximum likelihood estimates. It is hard for us to see a direct comparison between this study and our manipulations of sensory prediction error, but perhaps we did not fully appreciate the reviewers’ points.

3) Additional explanation is needed regarding the generalization. The learning model does not appear to explain the observed generalization effects (20 vs. 5 deg/s differences, and the gradual shift from linear to gaussian). Although this is addressed in the circuit model's predictions, it would be useful to find a way to validate either the descriptive or circuit model via simulations of the generalization behavior. In addition, the presentation should be improved to more clearly guide the reader through the logic of this section.

We assume responsibility for failing to explain the model’s generalization predictions clearly enough. As the reviewers point out, the cerebellar circuit model shows accurate simulations of the generalization effects for learning with pursuit target motion at 5 or 20 deg/s in Figure 12D and E, both of which were in the original version of the paper. We believe this comment referred to generalization effects predicted by the original conceptual model, which we have removed from the paper. We have addressed this comment further by reorganizing the presentation along the lines suggested by the reviewers, improving our explanations of the logic and the specifics of the generalization data, and linking the generalization function after 1000 trials more explicitly to how we modeled the inputs to the FTNs.

a) The first main model conclusion, which is that a slow learning system is required to explain the change in speed generalization to a linear relationship, is not explained well. Intuition is not provided for why the slow system produces a linear relationship when taught by the fast system which does not. When this modification to the model is introduced, multiple features are added (a second system, and feedback control of the error signal), and it isn't clear which of these, or both, are necessary to obtain linearity. Starting in the subsection “Strategy for testing the conceptual model of motor learning”, the authors explain the strategy saying "inputs", which are not illustrated in their conceptual model (Figure 5A). Readers, especially those who are not familiar with the cerebellar circuit, are left to wonder, because this strategy needs imagination based on a real circuit which has an eye velocity related-firing of input, plasticity site, input and plasticity-dependent output for expression of learned behavior. Readers may understand at last when they see Figure 10A, which is the cerebellar learning circuit.

We have taken several steps to address this concern. (1) We have reorganized the progression to introduce the cerebellar circuit architecture as Figure 1 and then to build up our computational model one step at a time as the data unfold. (2) We have added a new panel (Figure 8C) to explicitly show that the balance of learning and forgetting alone takes 1000-2000 trials for the emergence of a linear relationship between learning magnitude at instruction magnitude, much longer than the 100 trials in our data. (3) We have added inhibition of the climbing fiber input pathway as an explicitly separate step in our logic to show that this single addition causes the aforementioned relationship to be linear after 100 trials. (4) We have added the intuition that “Linearity emerges after 100 trials because recurrent feedback switches the mechanism driving asymptotic behavior from a tradeoff between learning and forgetting to an emergent property of the operation of a dynamic feedback loop”.

b) The description of the inference of input tuning when describing learning generalization is a bit obtuse. "The changing shape of the generalization function was further accentuated in Monkey RE after 2000 trials when probe trials expressed less behavioral learning (learning expression ratio <1.0) when the pursuit speed was faster in the probe than in the learning trial, suggesting Gaussian generalization.""Gaussian generalization" not defined anywhere. It should be, and the particular form used should be motivated. I assume what the authors mean is that it supports some kind of unimodal tuning of the input. Why can't the form be directly inferred from data, rather than assumed to be Gaussian? In general, the fitting to Equation 6 seems rather arbitrary.

We have revised the text surrounding this issue by abandoning the specifics of “Gaussian” and instead referring to the generalization function as “unimodal” based on the likelihood that any form of unimodal generalization would likely accurately describe our data. The crucial detail that our data suggests is that generalization peaks on the training pursuit speed after long bouts of learning (a learning expression ratio equal to 1.0). After many learning trials, the expression ratio is smaller than 1.0 for pursuit speeds both slower and faster than the trained speed in the learning trials. These data mandate a generalization function that peaks on the training speed and falls off as probe pursuit speeds deviate from the trained speed. A Gaussian generalization function fits this requirement (as does, for instance, cosine or von Mises generalization). Also, we have removed our analysis of the timescales of transition from a linear to a unimodal generalization due to the requirements of assuming a specific form for the unimodal function.

c) Regarding the generalization of learning to different probe speeds: There should be more discussion of this result from an algorithmic perspective and possible/alternative explanations. Can the results be explained by "confidence" about the speed during the learning trial and its relationship to the probe speed that increases over learning and, at the end of learning, results in the learning expression only generalizing to the same speed ("Gaussian" generalization)? Such a discussion could make more intuitive what is otherwise a difficult and technical section that leads to strange conclusion about the model (see next comment).

We agree that something about the exact design of the experiment is a factor that needs to be considered here. We knew that the alternation of instruction trials and probes might affect the outcome, but our pilot data using a 1:10 ratio of probes to instructions versus a 1:1 ratio suggested that the behavioral results were not dependent on the relative proportions of learning versus probe trials. In the Results, we have added an explicit point that the pattern of generalization depends not on whether the probe and learning speeds are the same or different, but rather on which speed is higher. This is a little harder to explain on the basis of “confidence”. We deal with a second issue related to generalization in the response to the next comment where we think out loud about other potential explanations for the same data, outside of differences in the functional properties of the signals that are subject to plasticity.

d) One of the hardest to swallow aspects of the model is the necessity of extremely different speed tuning of the inputs to the fast and slow systems. It is especially hard to swallow because these inputs both ultimately arise presumably from mossy fibers that contact nuclear cells directly or contact granule cells. In such a model, it would seem that either different mossy fibers for each system would be required, or some complicated transformation would need to happen at the granule cell. The authors should expand their discussion of how this might come about, and discuss whether there are other ways that the results might be explained without this strong assumption.

We understand the reviewers’ issues about inputs to FTNs that (1) are different from the inputs to granule cells and (2) are tuned in relation to eye speed. There is precedence in cerebellar literature for different “mossy fiber” inputs to related areas of the cerebellar cortex and deep cerebellar nuclei. For instance, areas that send afferent mossy fibers to the ventral paraflocculus, a region of the floccular complex that is crucial for smooth pursuit, send only sparse projections to the vestibular nucleus (Osanai et al., 1999; Nagao et al., 1997). Therefore, we feel reasonably comfortable with our modelling assumption that the inputs to FTNs versus the cerebellar cortex may be different.

However, we have zero neurophysiological evidence about the nature of the eye velocity inputs to FTNs. The general rule in the oculomotor system is that neural firing is linearly related to eye speed and eye position, rather than unimodally tuned. We now explain in the Discussion that there is little/no precedent for this prediction of the model.

Finally, we have tried to address this concern by changing the tenor of our narrative. We realize, and the paper now states clearly, that we have *chosen to model* the difference in generalization by assuming that plasticity operates on functionally different inputs to the two sites of learning. In no sense does this mean that the nervous system works the way the model does. We now state this clearly, even though we do not have a plausible alternative.

The finding about the transition from linear to unimodal generalization is both unexpected and unprecedented. Our model supports the principle that (1) there are two sites of learning and (2) learning works a bit differently at the two sites, and we suggest one way that might work. We think that the purpose of a model is to show how the brain *might* produce the data we have found, and to stimulate future research to address how the brain *actually* does so.

4) Results should be discussed in the context of previous work from other learning and memory systems. This paper highlighted four principles that explain learning of smooth eye responses that predict a change in target direction. Previous work on these principles in other learning and memory systems should be discussed.

Our Discussion already links the notion of transfer to a broader set of learning and memory systems. In addition, we link our conclusions about inhibition of learning in the cerebellar cortex with many other forms of motor learning that similarly show incomplete learning after extended training. Further, in the Results, we have extensively linked our analysis and model and conclusions to many other models based on data from other systems.

5) Several figures show data only from one monkey. Please justify this.

In the original manuscript, we had a few instances where we used data from Monkey RE as representative example data. However, all experiments were performed on both monkeys and summary results and statistics are always shown/performed for both monkeys. We have expanded the data presentation so that now only Figure 8D and Figure 9 show representative “data from one monkey”. The summary data for both of these exemplar data appear in Figure 8E and Figure 10C, respectively.